# Molecular Ancestry Across Allelic Variants of *SLC22A1*, *SLC22A2*, *SLC22A3*, *ABCB1*, *CYP2C8*, *CYP2C9*, and *CYP2C19* in Mexican-Mestizo DMT2 Patients

**DOI:** 10.3390/biomedicines13051156

**Published:** 2025-05-09

**Authors:** Adiel Ortega-Ayala, Carla González de la Cruz, Pedro Dorado, Fernanda Rodrigues-Soares, Fernando Castillo-Nájera, Adrián LLerena, Juan Molina-Guarneros

**Affiliations:** 1Department of Pharmacology, Faculty of Medicine, National Autonomous University of Mexico, Mexico City 04510, Mexico; ad.ortega@unam.mx; 2Personalised Medicine and Mental Health Unit, University Institute for Bio-Sanitary Research of Extremadura (INUBE), 06080 Badajoz, Spain; carla.gonzalezd@externos.salud-juntaex.es (C.G.d.l.C.); pdorado@unex.es (P.D.); fernanda.soares@uftm.edu.br (F.R.-S.); allerena@unex.es (A.L.); 3Faculty of Medicine and Health Sciences, University of Extremadura, 06006 Badajoz, Spain; 4Instituto de Ciências Biológicas e Naturais—ICBN, Universidade Federal do Triângulo Mineiro—UFTM, Uberaba 3838025-350, MG, Brazil; 5Centro de Salud T-III Portales, Servicios de Salud de la Ciudad de México, Ciudad de México 03660, Mexico; castillo_najera@facmed.unam.mx

**Keywords:** *SLC22* family, *CYP2C8*, *CYP2C9*, *CYP2C19*, antidiabetic drugs

## Abstract

**Background/Aims**: across protein-coding genes, single nucleotide allelic variants (SNVs) affect antidiabetic drug pharmacokinetics, thus contributing to interindividual variability in drug response. SNV frequencies vary across different populations. Studying ancestry proportions among SNV genotypes is particularly important for personalising diabetes mellitus type 2 (DMT2) treatment. **Methods**: a sample of 249 Mexican DMT2 patients was gathered. SNVs were determined through real-time PCR (RT-PCR). Molecular ancestries were determined as 3 clusters (Native-American, European, and African) based upon 90 ancestry markers (AIMS). Statistical inference tests were performed to analyse ancestry across 23 SNV genotypes. Allele and ancestry distributions were analysed through Spearman’s correlation. **Results**: ancestry medians were 65.48% Native-American (NATAM), 28.34% European (EUR), and 4.8% African (AFR). *CYP2C8*3* and *CYP2C8*4* were negatively correlated to NATAM, whereas positively to EUR. The activity score of *CYP2C9* was correlated to NATAM (Rho = 0.131, *p* = 0.042). *CYP2C19*17* and the activity score of *CYP2C19* were negatively correlated to NATAM. The correlation throughout *SLC22A1* variants, such as GAT in rs72552763, was positive by EUR, while A in rs594709 was negative thereby and positive by NATAM. *SLC22A3* variant C in rs2076828 was positively correlated to NATAM. NATAM patients present higher HbA1c levels with respect to Mestizo patients (*p* = 0.037). Uncontrolled patients (HbA1c ≥ 7%) have a larger NATAM ancestry (*p* = 0.018) and lower EUR (*p* = 0.022) as compared to controlled patients (HbA1c < 7%). **Conclusions**: there is a correlation between ancestry and some pharmacokinetically relevant alleles among Mexican DMT2 patients. Ethnicity is relevant for personalised medicine across different populations.

## 1. Introduction

Metformin is the first-line pharmacological treatment against diabetes mellitus type 2 (DMT2). Combined therapy is applied when therapeutic goals (HbA1c > 7%) are not achieved. Dual therapy drugs regarded as highly effective include pioglitazone, SGLT2i (sodium–glucose cotransporter 2 inhibitor), GLP-1 (glucagon-like peptide 1) antagonists, and sulphonylureas [1]. International Diabetes Federation data indicate that the largest DMT2 incidence in young people is reported among Canadian First Nations, Native-Americans, Indigenous Australians, and African-Americans; whilst the lowest incidence rates are found among Europeans and American Caucasians. These differences are explained through genetic predisposition, ethnicity, and healthcare access [2].

Notwithstanding the assortment of therapeutic options, DMT2 is characterised by a remarkable interindividual variability in terms of drug response, where approximately 30% is due to single nucleotide variants (SNVs). Genetic factors also play a relevant role in adverse reactions, as 59% of the most commonly cited drugs in adverse reaction surveys are metabolised by at least one enzyme carrying an altered-function allelic variant [3]. Metformin does not biotransform, and it mainly acts on the hepatocyte by inhibiting hepatic glucose production. Furthermore, it is transported across tissues by organic cation transporters (OCTs), primarily OCT1, OCT2, OCT-3, and P-glycoprotein (P-gp). These transporters are respectively coded by genes *SLC22A1*, *SLC22A2*, *SLC22A3*, and *ABCB1*, which are considered as highly polymorphic [4]. Drugs like sulphonylureas (SUs) and thiazolidinediones (TZDs) are metabolised by enzymes of cytochrome P-450 (CYP). SUs are principally metabolised by CYP2C9 and CYP2C19 [5], while TZDs are metabolised by CYP2C8 [6]. These CYP enzymes are coded by polymorphic genes whose phenotype can be defined in accordance with their genotype, resulting in ultra-rapid metaboliser (UM), poor metaboliser (PM), intermediate metaboliser, or normal metaboliser, which affect pharmacokinetics and pharmacodynamics of CYP-metabolised drugs [7,8]. Previous studies on Mexican populations have identified ancestry proportions and high frequencies of some allelic variants (rs9282541), even finding correlations with HDL (high-density lipoprotein) levels [9]; albeit, no ancestry or pharmacogenetic studies that we know of have been performed on Mexican DMT2 patients. In either CYPs or transporters, allelic variants can contribute to drug response variability. Several populations around the world present genetic variants even within the same continent [10], thus making the study and comprehension of inherent ancestral variability relevant for therapeutic recommendations against highly prevalent diseases of global incidence such as DMT2. The main aim of the present study was assessing Native-American, European, and African ancestry distribution across 23 allelic variants of 7 genes coding pharmacokinetic proteins involved with antidiabetic drugs such as sulphonylureas (*CYP2C9* and *CYP2C19*), thiazolidinediones (*CYP2C8*), and metformin (*SLC22A1*, *SLC22A2*, *SLC22A3*, and *ABCB1*). We have also explored the possible relationship between allelic variants and molecular ancestry; in particular, we explored the possible association between molecular ancestry and CYP enzymatic activity according to the activity score.

## 2. Material and Methods

### 2.1. Study Design

The present study is observational, not a clinical test. The study was conducted in accordance with the Declaration of Helsinki and was approved by the Research and Ethics Commissions of the Research Division of Universidad Nacional Autónoma de México’s Faculty of Medicine (Approval No. 105-2012). All patients provided written informed consent to participate in this study and no identifying information, including names, initials, date of birth, or hospital numbers, images, or statements are included in the manuscript. ‘Mestizo’ is primarily used to denote people of mixed European and non-European ancestry in the former Spanish Empire. Patient eligibility criteria were as follows: self-proclaimed Mexican-Mestizo ancestry of at least three generations [11].

### 2.2. Inclusion and Exclusion Criteria

Patients were recruited for the present study according to the following inclusion criteria: (i) the patient was undergoing either glibenclamide or metformin treatment, or a combination of the two; (ii) the patient had undergone a treatment schedule comprising a stable dose of these drugs for at least 3 months; (iii) the precedents and treatment characteristics of each individual was accessible via medical record at the corresponding healthcare centre, particularly, data concerning drug dosage (including hypoglycaemic agents) during the aforementioned 3-month period; and (iv) the medical file comprised anthropometric parameters and laboratory reports on a number of key biochemical variables (including HbA1c, fasting glucose levels, triglycerides, and cholesterol). The following exclusion criteria were taken into account: chronic alcoholism, previous pancreatic pathology, renal failure, hypoglycaemic treatment with insulin or insulin analogs, insufficient medical records, DMT1, and voluntary withdrawal.

### 2.3. Data Collection

Sample collection and clinical record reviewing were accomplished within a cohort of patients with DMT2 undergoing medical treatment and monitored at first-level public healthcare centres in Mexico City between July 2014 and October 2016.

### 2.4. Genotyping Procedure

The DNA was isolated and purified from blood samples using a QIAmp DNA extraction kit (Qiagen, Hilden, Germany). Genotyping for the different CYPs and transport variants (Appendix A: Analysed allelic variants with their respective PCR-TR probes across Mexican DMT2 patients (*n* = 248)) was performed using commercially available genomic DNA Taqman^®^ assays (Applied Biosystems, Foster City, CA, USA). Genotypes were assigned according to the presence of “key” SNVs associated with the relevant alleles (Appendix A). All assays included negative (no DNA) and positive (heterozygous and/or homozygous) control samples from previous studies of our group. Plates were read with an ABI 7300 real-time PCR system (Applied Biosystems, Foster City, CA, USA) and the following thermocycling conditions were applied for all assays: 10 min for initial denaturation at 95 °C, followed by 40 denaturation cycles of 15 s at 92 °C and annealing at 60 °C for 1 min. Allele discrimination lasted 30 s at 60 °C. Genotype-based phenotype predictions were assigned to *CYP2C8* and *CYP2C9* as follows: individuals carrying either two non-functional alleles or one non-functional allele plus a reduced function allele were poor metabolisers (gPMs), with an assigned activity score of 0–0.5. Intermediate metabolisers (gIMs) were those individuals with either two reduced function alleles or a normal function allele plus a reduced function or non-functional allele, with an activity score of 1–1.5. Normal metabolisers (gNMs) carry two normal function alleles, with an activity score of 2 [12]. There is no current activity score assignment consensus about CYP2C19; individuals are rather classified according to their *CYP2C19* genotype. However, we employed a classification of the different metaboliser phenotypes: poor metabolisers (gPMs) carry two non-functional alleles, activity score 0; intermediate metabolisers (gIMs) carry one normal function allele plus one non-functional allele, or an increased function allele plus one non-functional allele, activity score 1–1.5; normal metabolisers (gNMs) carry two normal function alleles, activity score 2; rapid (gRMs) and ultra-rapid metabolisers (gUMs) carry one normal function allele plus an increased function allele or two increased function alleles, respectively. These two latter groups are integrated into a single ultra-rapid metaboliser group (gUMs) with an activity score of >2 [7]. This phenotype group classification based on genotype and/or activity score has been developed from published CPIC Clinical Implementation Guides [7,12].

### 2.5. Genomic Ancestry Analysis

Individual genomic ancestry was determined in 238 individuals from Mexico. Out of these, 10 individuals did not register all 90 ancestry markers, thus no ancestry proportion was determined thereby. The African (AFR), European (EUR), and Native-American (NATAM) components were inferred by genotyping 90 ancestry informative markers (AIMs) from the same panel used in previous studies [13,14]. AIM genotyping was performed at the National Genotyping Centre (CEGEN) at Santiago de Compostela, Spain, using the Sequenom (San Diego, CA, USA) platform. Individuals from the three parental populations were also inserted in the final database: 114 Spaniards and 296 Peruvian Native-Americans from RIBEF-CEIBA [13], and 209 African Yoruba individuals from the 1000 Genomes Project [10]. The complete databases were transformed to .ped .map format using GLU 1.0b3 software (https://code.google.com/archive/p/glu-genetics/, accessed on 6 May 2025) and then analysed in the Admixture software Version 1.3.0 [15] in an unsupervised mode, assuming a tri-hybrid model (k = 3).

### 2.6. Statistical Analysis

This study’s statistical analyses and figures were obtained through R-4.2.0 [16] (available at: https://www.r-project.org/). The statistical analysis was carried out in 3 phases: (i) descriptive, (ii) inferential, and (iii) correlation.

#### 2.6.1. Descriptive Analysis

We observed the number and percentage frequencies of the analysed genotypes and alleles. In the case of *CYP2C8*, *CYP2C9*, and *CYP2C19*, the allelic frequencies tally referred to the activity score. The Hardy–Weinberg equilibrium was determined through Pearson’s Chi-squared test. The significant value was *p* < 0.05 (Appendix A: Allelic and genotypic frequencies and activity score distribution by *CYP2C8*, *CYP2C9*, and *CP2C19* within a sample of Mexican DMT2 patients (*n* = 248)).

#### 2.6.2. Inferential Analysis

We described ancestry proportions in terms of NATAM, EUR, and AFR, grouping genotypes of the relevant SNVs and the activity score of CYP2C8, CYP2C9, and CYP2C19. We conducted the corresponding Shapiro–Wilk and Kolmogorov–Smirnov tests. Statistical inference was performed across ancestry proportion genotypes through the Kruskal–Wallis test. For the post hoc analysis or those variants reported in 2 groups only, Mann–Whitney’s U test was applied. Post hoc test *p* value was adjusted following the Bonferroni and FDR (False Discovery Rate) methods for multiple comparisons. The significant value was *p* < 0.05.

#### 2.6.3. Correlation Analysis

Across statistically significant transporters and cytochromes (*CYP2C8*, *CYP2C9*, *CYP2C19*, rs72552763, rs594709, and rs2076828), Spearman’s rank correlation coefficient was determined by either the genotype’s allele or each individual’s diplotype (0, 0.5, and 1) and the ancestry proportion (NATAM, EUR, or AFR). The significant value was *p* < 0.05. Total minor alleles were tallied by individual, and Spearman correlation across ancestry proportions was performed using a scatter plot with a regression line at a 95% confidence interval adjusted by function lm() in stats (v4.2.0) through method = lm of geom_smooth in ggplot2 (v3.5.1). An analogue analysis was applied to the activity score of *CYP2C8*, *CYP2C9*, and *CYP2C19*.

#### 2.6.4. Clinical Biomarker and Ancestry Inference

Genetic ancestry classification (GA) was determined through ancestry proportion. Patients with a specific ancestry of >80% were assigned to the majority GA, whilst <80% patients were considered Mestizo [17]. HbA1c control was HbA1c < 7% and non-control was HbA1c ≥ 7% [1]. Statistical inference across GA and HbA1c control groups was performed using independent group inference tests. Shapiro–Wilk and Kolomogorov–Smirnov tests were performed as required. Statistical inference was carried out through Mann–Whitney’s U test and Pearson’s Chi-squared test, where *p* < 0.05 was statistically significant.

## 3. Results

### 3.1. Cytochrome and Transporter Allelic and Genotypic Frequencies

This study encompassed 248 DMT2 patients. Diplotype, allele, and activity score frequencies of *CYP2C8*, *CYP2C9*, and *CYP2C19* are summarised in Appendix A. OCT (*SLC22A1*, *SLC22A2*, and *SLC22A3*) and P-Gp efflux pump (*ABCB1*) allelic and genotypic frequencies are summarised in Appendix A. SNV allelic and genotypic frequencies in *SLC22A1*, *SLC22A2*, *SLC22A3*, and *ABCB1* (*n* = 248). The Hardy–Weinberg equilibrium revealed statistical significance by rs1045642 in *ABCB1* al (*p* = 0.028).

### 3.2. Ancestry Description

NATAM, EUR, and AFR ancestry was determined across 238 DMT2 Mexican patients. These ancestry proportions as well as ancestry determination clusters are displayed in Figure 1.

#### 3.2.1. Ancestry Inference Across *CYP2C8*, *CYP2C9*, and *CYP2C19*

The inferred ancestry proportion across the diplotypes and activity score of CYP2C8, CYP2C9, and CYP2C19 are summarised in Table 1.

##### *CYP2C8* Ancestry Inference

The corresponding diplotype analysis found statistical significance by CYP2C8 in NATAM (*p* = 0.001) and EUR (*p* = 0.009). We conducted a post hoc analysis for all three ancestry proportions grouped by *CYP2C8* diplotypes (Figure 2). Diplotype **1*/**1* was statistically different from genotypes **1*/**3* (p_Bonferroni_ = 0.029) and **1*/**4* (p_Bonferroni_ = 0.019) in NATAM (Figure 2A). The activity score (1.5 and 2) analysis found statistical significance in NATAM (p_Bonferroni_ = 0.00035), EUR (p_Bonferroni_ = 0.00036), and AFR (p_Bonferroni_ = 0.010) (Table 1 and Figure 2D–F).

##### *CYP2C9* Ancestry Inference

The inference of *CYP2C9* diplotypes revealed statistical significance in AFR (*p* = 0.019); however, the *p* value lost significance after Bonferroni and FDR adjustments.

The activity score analysis only found significance in AFR (*p* = 0.010). The post hoc analysis (Figure 3D–F) also found statistical significance among individuals reporting activity scores of 1.5 and 2 in NATAM (p_crude_ = 0.025), but it lost significance after the Bonferroni adjustment.

##### *CYP2C19* Ancestry Inference

The ancestry proportion assessment of *CYP2C19* diplotypes reported statistical significance in NATAM (*p* = 0.018), but neither in EUR (*p* = 0.110) nor in AFR (*p* = 0.140). However, a Bonferroni adjusted Mann–Whitney’s post hoc test did find significant differences in NATAM when comparing diplotypes **1*/**1* and **1*/**17* (p_Bonferroni_ = 0.018), and also between the latter and **1*/**2* (p_Bonferroni_ = 0.048) (Figure 4A). EUR and AFR ancestry reported no statistical significance by *CYP2C19* diplotypes (Figure 4B,C).

The ancestry distribution assessment across the different CYP2C19 metabolising phenotypes (PM: poor metaboliser, UM: ultra-rapid metaboliser, 1, 1.5, and 2: normal metaboliser) revealed significance in NATAM (*p* = 0.041). The CYP2C19 activity score inference analysis (Figure 4) found statistical differences only between 1 vs. UM (p_Bonferroni_ = 0.048) and 2 vs. UM (p_Bonferroni_ = 0.018) in NATAM (Figure 4D).

#### 3.2.2. Ancestry Inference in Transporter SNVs

Ancestry proportion was assessed across the different genotypes of OCT1, OCT2, and OCT3 coded by *SLC22A1*, *SLC22A2*, and *SLC22A3* (Table 2). We also assessed ancestry distribution across the variants of the efflux pump P-gp coded by *ABCB1*. We observed no significant differences between the three different ancestry proportions in the three different genotypes of rs72552763 in *SLC22A1*. However, the post hoc test found significance between GAT/GAT and del/del in NATAM (p_crude_= 0.030), which was not the case in EUR and AFR (Figure 5A–C); albeit, the *p* value did not remain significant after Bonferroni and FDR adjustments. An analysis of rs594709, another variant of *SLC22A1*, found differences in NATAM (*p* = 0.008) and EUR (*p* = 0.020). The post hoc test reported significance by wild-type (AA) and minor allele (AG, GG) carriers in NATAM (p_crudeAA vs. AG_ = 0.026 and p_crudeAA vs. GG_ = 0.024) and EUR (p_crudeAA vs. AG_ = 0.043 and p_crudeAA vs. GG_ = 0.040), but this significance was lost after Bonferroni and FDR adjustments (Figure 5D,E). In the case of rs316019 in *SLC22A2*, we found differences between genotypes CC and CA in NATAM (*p* = 0.009) and EUR (*p* = 0.005), but not in AFR (*p* = 0.831).

The ancestry distribution assessment across the genotypes of rs2076828 in *SLC22A3* reported statistical differences in NATAM (*p* < 0.001), EUR (*p* < 0.001), and AFR (*p* = 0.609) between CC, CG, and GG. The Bonferroni adjustment post hoc tests revealed the ancestry proportion to be different among rs2076828 genotypes, except for AFR (Figure 5G,H).

The ancestry proportion analysis found no significant values in either NATAM, EUR, or AFR by any genotype of rs2032582, rs1128503, or rs105642 in *ABCB1*.

### 3.3. Correlation Analysis

In the search for a correlation between ancestry and alleles, we performed Spearman correlation models for those cases where the inference analysis had reported statistical significances by the relevant cytochromes and transporters (CYP2C8, CYP2C9, CYP2C19, rs72552763 in *SLC22A1*, and rs2076828 in *SLC22A3*).

#### 3.3.1. Correlation Analysis for *CYP2C8*

The assessment reported *wt* in correlation with NATAM (Rho = 0.250, *p* < 0.001), EUR (Rho = −0.207, *p* = 0.001), and AFR (Rho = −0.165, *p* = 0.010); *3 in correlation with NATAM (Rho = −0.178, *p* = 0.006), EUR (Rho = 0.162, *p* = 0.012), and AFR (Rho = 0.102, *p* = 0.117); *4 in correlation with NATAM (Rho = −0.194, *p* = 0.002), EUR (Rho = 0.149, *p* = 0.023), and AFR (Rho = 0.146, *p* = 0.023) (Table 3).

Our analyses suggest that allele *CYP2C8*1* is positively correlated to NATAM whilst negatively correlated to EUR and AFR ancestries. Alleles **3* and **4* appear negatively correlated to NATAM whilst negatively correlated to EUR; only **4* was positively correlated to AFR.

#### 3.3.2. Correlation Analysis for *CYP2C9*

The assessment reported *wt* in correlation with NATAM (Rho = 0.135, *p* = 0.036); **2* in correlation with NATAM (Rho = −0.133, *p* = 0.039), and AFR (Rho = 0.171, *p* = 0.007); **3* reported no correlation with any ancestry proportion. The activity score was in correlation with NATAM (Rho = 0.131, *p* = 0.042). These results suggest a positive correlation between allele *wt* of *CYP2C9* and NATAM’s activity score (Table 4).

#### 3.3.3. Correlation Analysis for *CYP2C19*

The Spearman assessment reported *wt* in negative correlation with EUR (Rho = −0.096, *p* = 0.026); **17* in negative correlation with NATAM (Rho = −0.198, *p* = 0.002), and positive correlation with EUR (Rho = 0.162, *p* = 0.011) and AFR (Rho = 0.137, *p* = 0.001). The activity score analysis reported correlation only with NATAM (Rho = −0.132, *p* = 0.041). We observed no correlation between **2* and any ancestry proportion. It was not possible to assess **4* due to the amount of this allele’s carriers within the sample (*n* = 1) (Table 5).

#### 3.3.4. Correlation Analysis for Organic Cation Transporters (OCTs)

The assessment reported allele GAT of rs72552763 in *SLC22A1* in positive correlation with EUR (Rho = 0.132, *p* = 0.041); A of rs594709 in *SLC22A1* in positive correlation with NATAM (Rho = 0.177, *p* = 0.005) and negative correlation with EUR (Rho = −0.162, *p* = 0.012); C of rs2076828 in *SLC22A3* in positive correlation with NATAM (Rho = 0.289, *p* < 0.001) and negative correlation with EUR (Rho = −0.258, *p* < 0.001). Since rs72552763, rs594709, and rs2076828 have only two alleles, the minor allele analysis differs only in correlation coefficient sign (Table 6).

#### 3.3.5. Correlation Analysis for Carried Allele Tally

We searched for a correlation between ancestry and the number of minor alleles by tallying these for each individual. This assessment found no correlation with NATAM (*p* = 0.684), EUR (*p* = 0.871), or AFR (*p* = 0.696) (Appendix A. Transporters (*SLC22A1*, *SLC22A2*, *SLC22A3* and *ABCB1*). A similar analysis was conducted for the minor alleles among the cytochromes (Appendix A. Cytochromes *CYP2C8*, *CYP2C9*, and *CYP2C19*). Therein, we observed a correlation with NATAM (Rho = −0.173, *p* = 0.007) and EUR (Rho = 0.164, *p* = 0.011), but not with AFR (Rho = 0.077, *p* = 0.235).

#### 3.3.6. Correlation Analysis for Activity Score and Ancestry Proportion

In the search for a correlation between the activity score of *CYP2C8*, *CYP2C9*, and *CYP2C19*, we generated scatter plot graphics representing an adjusted regression line (Figure 6). The Spearman analysis of *CYP2C8* reported a correlation between the activity score and NATAM (Rho = 0.250, *p* = 0.0001), EUR (Rho = −0.207, *p* = 0.001), and AFR (Rho = −0.165, *p* = 0.010). These numbers suggest that, within our sample, a larger proportion of NATAM is correlated with an increased activity score by CYP2C8 and, inversely, preponderant proportions of EUR and AFR are correlated with a reduced activity score. Our results also describe a positive correlation between the activity score and NATAM (Rho = 0.131, *p* = 0.042) by CYP2C9. For *CYP2C19*, we observed a negative correlation between the activity score and NATAM (Rho = −0.132, *p* = 0.041).

### 3.4. Clinical Biomarker and Ancestry Inference

HbA1c levels by NATAM (*n* = 37) were 7.33 (6.72–9.93), while Mestizo patients reported 6.86 (6.10–8.58), *p* = 0.037. We found statistical significance for other biomarkers such as size, weight, BMI, and rs2076828 frequency when comparing NATAM and Mestizo patients (Table 7).

The HbA1c level inference (Table 8) revealed that controlled patients (HbA1c < 7%) had a median NATAM ancestry of 64.18, while non-controlled patients (HbA1c ≥ 7%) had 67.16 (*p* = 0.018). Similar results were observed by EUR (*p* = 0.022).

## 4. Discussion

### 4.1. American Population Ancestry

The ethnic identity of most American countries originated during the 16th and 17th centuries AD (Anno Domini) through the interaction between indigenous, European, and African populations [18]. In this historical context and also in accordance with our own expectations, our results indicate that within this DMT2 sample, the larger ancestry proportion is NATAM (65.48%), followed by EUR (28.34%) and AFR (<5%).

These results are similar to previous reports on American samples, such as the African-Ecuadorian, the Montubio and Tsáchila Mestizo [18]; Lima, Peru [19]; and Mexico City [9]. However, our results differ from studies on Colombian [20,21] and Brazilian samples [22], which reported a larger European proportion (47.7–74.6%), whilst a study on a Dominican population [14] reported a highly non-European admixture with a significant African element. It shall be mentioned that these previously studied samples from Mexico, Colombia, Ecuador, Peru, and Dominican Republic recruited participants regardless of their health, whilst the Brazilian study was conducted on DMT1 patients from the Federal University of Maranhão Hospital. Our own sample was constituted of DMT2 patients from two different Mexico City Health Centres. These similarities between our sample and Ecuadorian and Peruvian populations, as well as the differences with respect to Brazil, Colombia, and Dominican Republic, might be explained by a larger native population at the time of the European contact, as was the case of the Mexica Empire in Mexico and the Inca Empire in Andean regions in Peru, Ecuador, and Bolivia [20].

### 4.2. Ancestry and Pharmacogenetics

This study is the first report on genetic ancestry across relevant pharmacogenetic allelic variants within a sample of DMT2 Mexican patients.

#### 4.2.1. *CYP2C8*

We found statistical differences in Native-American (NATAM) and European (EUR) ancestry proportions by the *CYP2C8* diplotype, where NATAM ancestry was larger among allele **1* carriers and lesser among alleles **3* and **4* (Table 3). Our analyses suggest that *CYP2C8*1* is positively correlated to NATAM whilst negatively correlated to EUR and AFR ancestries. Alleles **3* and **4* appear negatively correlated to NATAM whilst negatively correlated to EUR; only **4* was positively correlated to AFR. Treatment individualisation recommended by the American Diabetes Association (ADA) and the European Association for the Study of Diabetes (EASD) focuses on thiazolidinediones (TZD) such as pioglitazone and rosiglitazone [6]. These drugs (TZD) are widely metabolised by *CYP2C8*. According to the 1000 Genomes database, the frequency of *CYP2C8*3* is approximately observed in 8–15% of EUR ancestry populations [6,23], whilst being significantly lower (~2%) among Peruvian individuals [23]. The frequency of *CYP2C8*3* in our present study amounts to 4%.

Pharmacokinetic studies among healthy individuals have proven that *CYP2C8*3* carriers report a meek TZD metabolism increase and 36% less rosiglitazone plasmatic concentration [6]. This discreet pharmacokinetic difference may result in a relevant pharmacodynamic effect, as a reduced drug exposure among *CYP2C8*3* carriers has resulted in a lower HbA1c reduction as well as a lower weight increase when compared to **1*/**1* [6]. Our results allow us to infer TZD might be a viable therapeutic alternative for Mexican-Mestizo and other significantly NATAM-ancestry patients would likely report greater benefits when compared to EUR, albeit NATAM could also imply an increase in TZD adverse reactions. While there is currently no clinical recommendation guide on *CYP2C8*-metabolised drugs, this enzyme has been described along with *CYP2C9* in nonsteroidal anti-inflammatory drugs (NSAIDs) guidelines [8]. Multiple studies have proven a linkage disequilibrium between alleles *CYP2C8*3* and *CYP2C9*2* across different populations [24,25]. Carriers of both alleles have reported a significantly different ibuprofen metabolic profile, which is subjected to an inferior clearance value and a larger area under the curve (AUC) as compared to the wild-type [26,27]. A classification based on the *CYP2C8* phenotype has been recently proposed: ultra-rapid metabolisers (UMs) for **3*/**3*, rapid metabolisers (RMs) for **1*/**3*, normal metabolisers (NMs) for **1*/**1*, intermediate metabolisers (IMs) for **1*/**4*, and poor metabolisers (PMs) for **4*/**4* [28]. For CYP2C8, however, we chose an activity score classification just like the existing one for CYP2C9 [12], because the effect of *CYP2C8*3* has been observed as dependent on the specific substrate that can provoke both increased or decreased enzymatic activity [28], and healthy individuals carrying this allele have reported altered enzymatic activity in antidiabetic drug metabolism as for rosiglitazone [6]. *CYP2C8*4* is generally associated with an enzymatic activity reduction by drugs such as montelukast [29]. While limitations must be presently acknowledged regarding UMs and PMs, this might be the start of an activity score assignment for *CYP2C8* in future diabetes treatments.

#### 4.2.2. *CYP2C9*

In the Americas, its frequency has been reported as 2.4% amongst Peruvians from Lima and 13.9% in Puerto Rico [23]. The frequencies of *CYP2C9* variants observed in this study are in accordance with previous studies in Mexican Americans [30], Mexican Mestizos [31], and different indigenous populations from Mexico such as Tepehuanos, Seris, Mayos, Tarahumaras, Mayos, Guarijíos, Huicholes, and Coras [32,33,34,35]. These studies have reported low frequencies of *CYP2C9*2* as well as low frequencies or even absence of *CYP2C9*3*. The allelic distribution of the *CYP2C9* gene in this Mexican population is compatible with the genomic assembly of the constitutive ethnic origin of this Hispanic group, an admixture of ancient Indians living in Central and North America and White Europeans coming mostly from Spain [31]. This result, we hereby report, suggests there are likely no substantial differences across *CYP2C9* allelic frequencies between healthy individuals and DMT2 patients in Mexican populations. Likewise, *CYP2C9*3* amounts to 5.6% amongst Finns and up to 8.4% amongst Spaniards, whilst Peruvians from Lima report a frequency of 1.2% and Colombians from Medellín report 6.4%. Within our sample, *CYP2C9*3* frequency was 2.82% (Table 1). The frequencies we observed are similar to those previously reported by Sánchez-Pozos et al., 2016 [36], who studied an array of Mexican indigenous populations. They found that 100% of the Mazahua carry the diplotype **1*/**1*, and a near-overall absence of alleles **2* and **3*, where the Chontal reported the highest frequencies of *CYP2C9*2* (0.017%) and *CYP2C9*3* (0.05%). The ancestry and diplotype inference analysis, as well as the activity score of CYP2C9 (Table 4), found no statistical differences across NATAM, EUR, or AFR. However, the correlation analysis (Table 7) revealed positive correlations between *CYP2C9*1* and NATAM, as well as NATAM and the activity score of CYP2C9. The correlation was negative between NATAM and *CYP2C9*2*. These results line up with the frequencies reported by the 1000 Genomes project, given that *CYP2C9*2* is slightly higher among EUR, as is the case in Spain and Italy [23], and also American populations with larger EUR ancestry such as Medellín, Colombia [20,21].

According to Clinical Pharmacogenetics Implementation Consortium (CPIC) guidelines, the enzymatic activity of *CYP2C9* and the subsequent classification of an individual as a normal (**1*/**1*), intermediate (**1*/**2*, **1*/**3*, **2*/**2*), or poor (**2*/**3*, **3*/**3*) metaboliser depends on the presence of alleles **1*, **2*, and **3*, and it is particularly relevant for metabolising NSAIDs, phenytoin, and warfarin [8]. Sulphonylureas (SUs) too are a group of drugs metabolised by *CYP2C9* and constitute combined therapy options along with metformin when no adequate glycaemic control is achieved [1]. The most recent meta-analysis performed on DMT2 patients has revealed that *CYP2C9*2* carriers are at a higher hypoglycaemic episode risk as a theoretical consequence of increasing the exposure of a reduced-activity enzyme to SUs [37].

*CYP2C9*2* and *CYP2C9*3* frequencies are very low amongst Mexican indigenous populations [34,35,36] and Mexican DMT2 patients [38,39]. Studies on SU-treated patients suggest that *CYP2C9*1*/**3* carriers achieve a better glycaemic control as compared to *CYP2C9*1*/**1* wild-type homozygous carriers [40]. Therefore, patients with a larger EUR ancestry are likely to benefit the most by SU treatment as compared to more significant NATAM proportions, because it is among these patients that *CYP2C9*2* and *CYP2C9*3* frequencies are below 1%. This result might offer an explanation as to why 88.9% of Mexican DMT2 patients treated with a combination of metformin + SUs are not under glycaemic control criteria [39].

#### 4.2.3. *CYP2C19*

SUs are primarily metabolised by *CYP2C9* and, to a lesser extent, *CYP2C19* [5]. According to CPIC guidelines on clopidogrel, voriconazole, selective serotonin reuptake inhibitors, tricyclic antidepressants, and proton-pump inhibitors, **17* is considered as an increased-function allele, the **1*/**1* diplotype is a normal-function allele, and the rest (**2*, **3*, **9*, **12*, **14*) are reduced-function alleles [7].

According to previous studies, reduced-function alleles *CYP2C19*2* and *CYP2C19*3* are common amongst East Asian populations, which respectively carry 31% and 6.7% as compared to Caucasian (18%, <0.1%) and American populations (10%, <0.1%) [41]. So far, the highest *CYP2C19*17* frequency has been reported among EUR (22.4%) and AFR (23.5%), whilst American individuals have reported 12.0% [41]. We have found no differences between NATAM, EUR, and AFR across the different *CYP2C19* genotypes (Table 4). Moreover, the correlation analysis (Table 8) revealed a negative correlation by *CYP2C19*17*-NATAM, while *CYP2C19*17* was positive in EUR and AFR; these results match the previous studies by Zhou et al., 2017 [41], and the 1000 Genomes project, where *CYP2C19*17* distribution was 4.1% among Peruvians, 21.5% among Spaniards, and 23.5% in different African Countries such as Nigeria, Gambia, Kenya, and others [23]. A study performed on healthy Chinese males suggests that *CYP2C19* plays a more relevant role than *CYP2C9* in gliclazide metabolism [42], albeit no *CYP2C19*17* carriers were reported therein [43]. Nevertheless, this study also suggested *CYP2C19* may involve enhanced glipizide efficacy and adverse reaction minimisation [43].

While the effect of *CYP2C19*17* on SUs has been scarcely studied so far, reports on Asian populations allow us to infer a considerable *CYP2C19* influence on SU’s therapeutic efficacy. Considering its increased enzymatic activity when interacting with other drugs such as clopidogrel [7], *CYP2C19*17* may withhold lower SU concentrations, which would imply reduced effects and adverse reactions. However, these phenomena could be subjected to *CYP2C9* genotypic variants [43]; therefore, even if we have found the aforementioned ancestry correlation, it is necessary to further study SU pharmacokinetics and the influence of both *CYP2C9* and *CYP2C19*.

#### 4.2.4. OCT Transporters

Pharmacological response to metformin significantly varies across individuals. Some studies indicate that metformin treatment does not achieve its therapeutic goals in up to 35% of the intaking patients, thus calling for a shift to combined therapy [4]. Through an ancestry proportion analysis across several *SLC22A1*, *SLC22A2*, *SLC22A3*, and *ABCB1* variants, we found statistical differences for rs594709 and rs628031 in *SLC22A1*, rs316019 in *SLC22A2*, and rs2076828 in *SLC22A3* (Table 4). We found a positive correlation between *SLC22A1* GAT in rs72552763 and EUR (Table 8). An ancestry proportion comparison throughout rs72552763 genotypes revealed that del/del carriers report a larger NATAM ancestry as compared to GAT/GAT (Figure 5A). Inversely, GAT/GAT reported a larger EUR ancestry as compared to del/del (Figure 5B). These results are consistent with the 1000 Genomes project, where Peruvians from Lima (the most akin population regarding Mexicans in this project) reported a minor del allele frequency of 37.6% (31.45% in our sample), whilst European populations such as Spain and Italy reported 16.4% and 19.6%, respectively [23].

A study on metformin’s steady state in diabetic patients found that its concentration is lower amongst minor allele rs72552763 *SLC22A1* carriers [44]. OCT1 is the most relevant transporter vis-a-vis metformin’s hepatic capture and, as proven by Christensen, del is a reduced-function allele transporting less metformin into the hepatocyte, as human-based studies have reported. In these surveys del/del carriers have presented a lower hepatic volume of metformin distribution as compared to GAT/GAT [45]. This means that metformin’s therapeutic efficacy could be lower among these patients, as previously reported in Mexican DMT2 patients undergoing metformin treatment [46,47].

If a large NATAM ancestry proportion implies the double deletion in rs72552763 *SLC22A1*, we may assume this consequential impact on metformin treatment. In addition, we found a positive correlation between NATAM and allele A by *SLC22A1*rs594709 (Table 8). Our results suggest that AA carriers share a larger NATAM proportion with respect to GG and AG (Figure 5D–F). According to the 1000 Genomes project, A was found in 73.6% of Peruvians; in our sample, it was 85.69%. European individuals reported 58.7% (Dyer et al., 2025 [23]). In the case of *SLC22A1* rs594709, Mexican DMT2 patients have reported no differences in HbA1c decrease across AA, AG, and GG undertaking metformin over a 12-month period. However, the same study included an analysis adjustment according to sex, disease duration, and abdominal circumference, which revealed higher HbA1c levels by GG [48].

Another study on Chinese diabetic patients found no differences in HbA1c levels when comparing variants through a recessive genotypic model (AA + AG vs. GG), but it did report a more significant fasting glucose decrease by AA in rs594709 and AA in rs2289669 of *SLC47A1* as compared to AA in rs594709 and G in rs2289669 [49]. These data lead us to suppose that rs594709 plays no relevant role by itself regarding metformin’s therapeutic efficacy and that the correlation between ancestry and rs594709 alleles has no clinical pharmacogenetic relevance either. Another variant by which we observed statistical differences regarding ancestry was rs316019 of *SLC22A2*, an OCT2-coding gene expressed on the basolateral membrane, and whose main function is metformin urinary excretion along MATE2, coded by *SLC47A2* [50].

Our results suggest that CC carriers have a discretely larger NATAM proportion. According to the 1000 Genomes project, A is present in 5.3% of Peruvians, whilst our sample reported a 4.8% frequency. EUR has reported 8–9% and AFR 1.3% [23]. The frequency of A is very similar throughout the three ancestries. Moreover, a cell study has proven metformin uptake is similar by both the wild-type genotype and the variant genotype; thus, any difference in metformin uptake might be clinically scarce or insignificant [51].

For rs2076828 in *SLC22A3*, an OCT3-coding gene, we found a positive correlation between C and NATAM, in contrast with EUR, for which we observed a negative correlation. Thereby, we also observed that a larger EUR proportion entails a higher G frequency. The 1000 Genomes project reported a 15.9% G frequency among Peruvians while our study recorded 15.52%, in contrast with European populations where it rises up to 50%; thus, our results are consistent with previous reports [23]. This variant has been studied in knockout mice, where the distribution volume and metformin elimination were lower among knockout (variant carriers) individuals. Furthermore, oral bioavailability was lower when compared to wild-type carriers, thereby implying a lesser metformin effect among G carriers [52]. Studies on 69 Mexican patients undergoing metformin treatment found no association between control time and the variant’s presence [46]. The studies hereby mentioned are, to the best of our knowledge, the only ones performed on humans where the effect of rs2076828 in *SLC22A3* on metformin monotherapy response is evaluated. In view of the significant differences throughout ancestry proportions, which are consistent with the 1000 Genomes project, we believe further metformin response studies on this variant would be very relevant. The frequency difference by G’s minor allele represents the largest breach between Peruvians, Mexicans, and Caucasians, therefore proving metformin response differences would establish rs2076828 in *SLC22A3* as a clear pharmacogenetic biomarker applicable to more than one population.

### 4.3. Ancestry, Clinical, and Pharmacogenetic Biomarkers

To the best of our knowledge, this is the first report on ancestry impact over HbA1c levels among Mexican DMT2 patients. We are hereby reporting that NATAM patients (>80% ancestry) present higher HbA1c levels as compared to Mestizo patients. This result matches a previous study whereby high-proportion NATAM individuals (>50% ancestry) reported low-functioning β pancreatic cells [20], which might lead to lower insulin levels and higher glucose and HbA1c levels.

Other studies have reported associations between DMT2 and non-EUR ancestry among Latino populations [53,54].

We also observed that NATAM patients present the highest HbA1c levels in spite of having the lowest BMI as compared to Mestizo individuals. This result matches a previous study on a Mexican cohort of Houston, Texas residents [55] as opposed to eight contiguous indigenous reservations of North America [56].

While we found a correlation between some SNV enzymes and transporters with NATAM, EUR, and AFR, we observed no differences across these variants’ frequencies when comparing NATAM and Mestizo, or control (HbA1c < 7%) and non-control (HbA1c ≥ 7%). This might be due to our sample’s recruiting criteria (DMT2 without specific treatment). This is the most evident limitation of our work, because a variant may be not observed if the diagnosed drug is not transported or metabolised by the transporter or enzyme of a specific variant.

## 5. Conclusions

These results remark ancestry’s relevance for DMT2’s clinical pharmacogenetics. By identifying differences throughout genotypes and detecting a correlation between SNV distribution and pharmacokinetic enzymatic activity by antidiabetic drugs such as metformin, TDZs, and SUs, this study highlights the importance of a personalised medical approach to DMT2. The most relevant variants found within Mexican populations are *CYP2C8*3* (rs11572080), *CYP2C8*4* (rs1058930), *CYP2C9*2* (rs1799853), *CYP2C9*3* (rs1057910), *CYP2C19*17* (rs12248560), rs72552763, and rs594709 in *SLC22A1*, and rs2076828 in *SLC22A3*. These results enhance our comprehension of therapeutic efficacy and antidiabetic drug safety across different populations whose ancestry proportion has been molecularly identified. Further studying ancestry influence and high-prevalence disease pharmacogenetics is a crucial step towards personalised medicine.

## Figures and Tables

**Figure 1 biomedicines-13-01156-f001:**
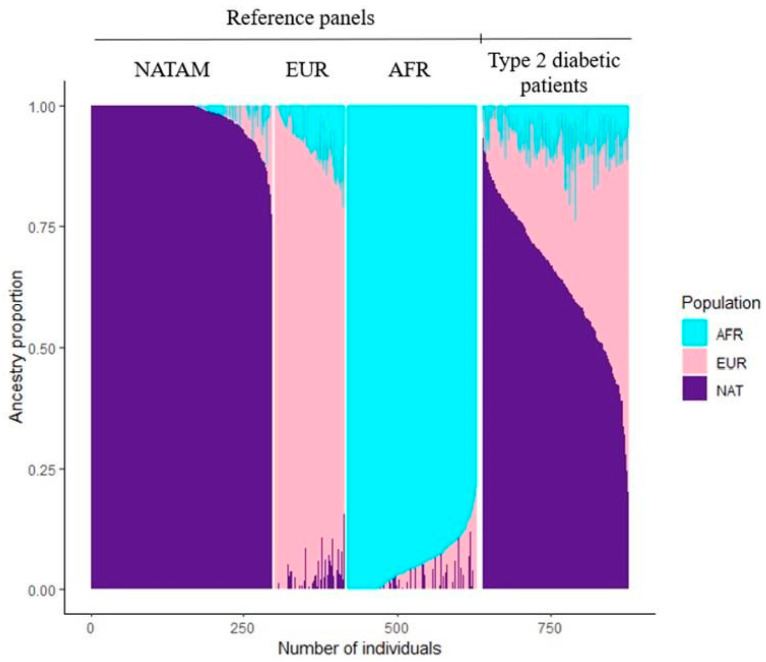
Reference panels and resulting clusters by using ADMIXTURE among Mexican DMT2 patients (*n* = 238).

**Figure 2 biomedicines-13-01156-f002:**
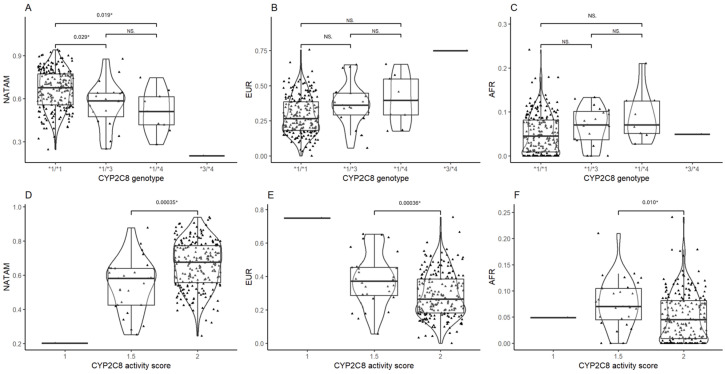
Ancestry proportion comparison (NATAM, EUR, and AFR) across our DMT2 patient sample, grouped by genotype and CYP2C8 activity score. Panels (**A**–**C**) are genotype groupings (**1*/**1*, **1*/**3*, **1*/**4*, and **3*/**4*). Panels (**D**–**F**) are activity score groupings (1, 1.5, and 2). The *p* value corresponds to Bonferroni’s post hoc adjustment test. * *p* < 0.05, NS.: not significant.

**Figure 3 biomedicines-13-01156-f003:**
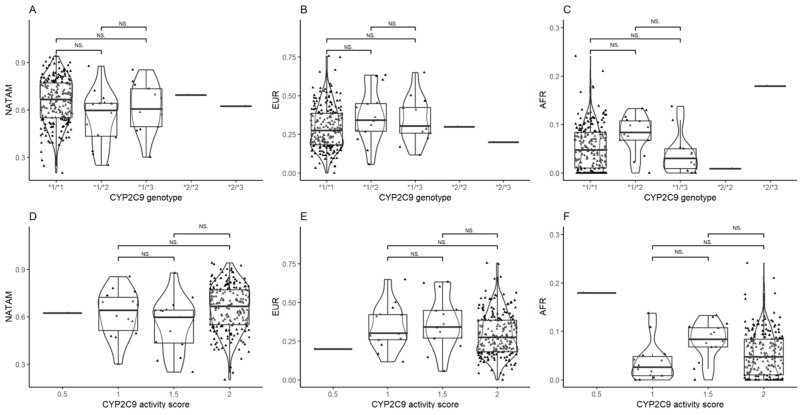
Ancestry proportion comparison (NATAM, EUR, and AFR) across our DMT2 patient sample, grouped by genotype and CYP2C9 activity score. Panels (**A**–**C**) are genotype groupings (*1/*1, *1/*2, *1/*3, *2/*2, and *2/*3). Panels (**D**–**F**) are activity score groupings (0.5, 1, 1.5, and 2). The *p* value corresponds to Bonferroni’s post hoc adjustment test. NS.: not significant.

**Figure 4 biomedicines-13-01156-f004:**
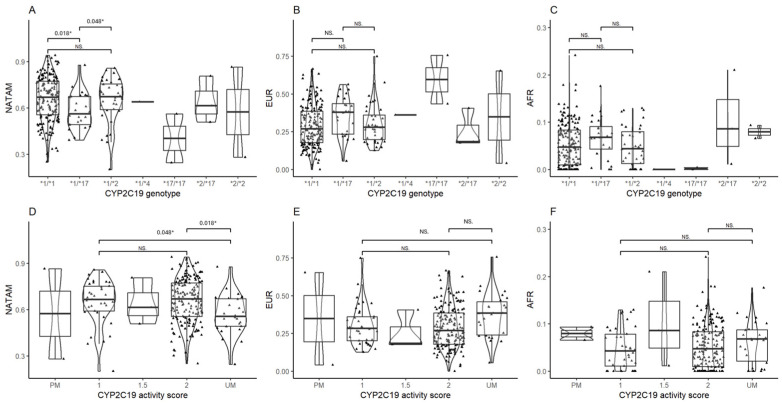
Ancestry proportion comparison (NATAM, EUR, and AFR) across our DMT2 patient sample, grouped by genotype and CYP2C19 activity score. Panels (**A**–**C**) are genotype groupings (*1/*1, *1/*17, *1/*2, *17/*17, *2/*17, and *2/*2). Panels (**D**–**F**) are activity score groupings (PM, 1, 1.5, 2, and UM). The *p* value corresponds to Bonferroni’s post hoc adjustment test. * *p* < 0.05, NS.: not significant.

**Figure 5 biomedicines-13-01156-f005:**
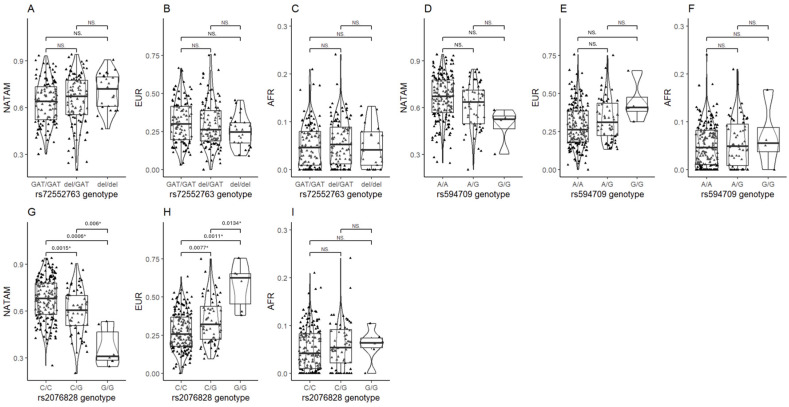
Ancestry proportion comparison (NATAM, EUR, and AFR) across our DMT2 patient sample, grouped by OCT variant genotypes. Panels (**A**–**C**) show genotype groupings of rs72552763 in *SLC22A1* (GAT/GAT, GAT/del, and del/del). Panels (**D**–**F**) show genotype groupings of rs594709 in *SLC22A1* (C/C, A/G, and G/G). Panels (**G**–**I**) show genotype groupings of rs2076828 in *SLC22A3* (C/C, C/G, and G/G). The *p* value corresponds to Bonferroni’s post hoc adjustment test. * *p* < 0.05, NS.: not significant.

**Figure 6 biomedicines-13-01156-f006:**
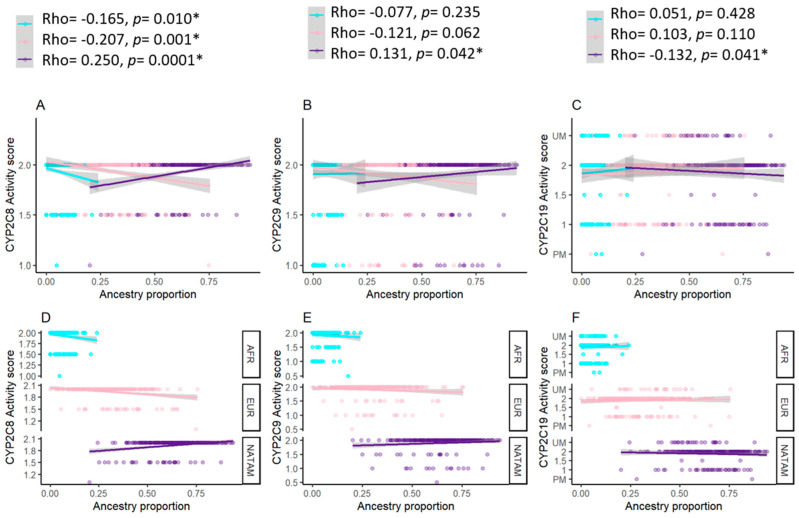
Activity score of cytochromes coded by *CYP2C8*, *CYP2C9*, and *CYP2C19*, hereby grouped according to ancestry proportion (NATAM, EUR, and AFR). Panels (**A**–**C**) show the relevant points and the adjusted by linear regression in accordance with ancestry. The path represents the adjusted regression line for ancestry proportion and activity score of each cytochrome in a linear model. The shaded areas represent the line’s 95% confidence interval. Rho corresponds to Spearman’s correlation coefficient. Panels (**D**–**F**) show the relevant points and the adjusted regression line’s facets in accordance with ancestry. Blue dots correspond to AFR ancestry, pink dots correspond to EUR ancestry, and purple dots correspond to NATAM ancestry. * *p* < 0.05.

**Table 1 biomedicines-13-01156-t001:** Ancestry distribution percentage across activity score and genotypes of *CYP2C8*, *CYP2C9*, and *CYP2C19* among Mexican DMT2 patients (*n* = 238).

	NATAM	EUR	AFR	Activity Score	NATAM	EUR	AFR
*CYP2C8*							
**1*/**1*	67.56 (55.72–77.34)	26.42 (18.20–38.50)	4.49 (0.90–8.18)	1	-	-	-
**1*/**3*	58.46 (47.42–63.94)	36.11 (29.17–44.69)	6.98 (3.67–10.10)	1.5	58.07 (42.54–63.91)	37.05 (28.69–45.36)	6.99 (4.47–10.47)
**1*/**4*	51.06 (41.81–61.34)	39.48 (29.13–54.80)	6.99 (5.09–12.46)	2	67.56 (55.72–77.34)	26.42 (18.20–38.50)	4.49 (0.09–8.18)
**3*/**4*	-	-	-				
*p ^KW^*	0.001 *	0.009 *	0.062	*p* ^U^	<0.001 **	0.003 *	0.035 *
*CYP2C9*							
**1*/**1*	66.52 (55.08–77.17)	27.34 (18.20–38.59)	4.71 (1.00–8.39)	1	64.05 (51.32–72.40)	30.07 (25.95–42.20)	2.57 (0.85–4.80)
**1*/**2*	56.45 (44.83–64.32)	34.00 (26.99–44.88)	8.31 (6.76–10.76)	1.5	56.45 (43.33–64.32)	34.00 (26.99–44.88)	8.31 (6.76–10.76)
**1*/**3*	60.54 (49.38–73.31)	30.34 (25.71–42.29)	2.95 (0.91–5.02)	2	66.52 (55.08–77.17)	27.34 (18.20–38.59)	4.71 (1.00–8.39)
**2*/**2*	-	-	-				
**2*/**3*	-	-	-				
*p ^KW^*	0.061	0.290	0.019 *	*p* ^KW^	0.137	0.175	0.010 *
*CYP2C19*							
**1*/**1*	66.81 (55.54–77.35)	26.85 (17.76–38.62)	4.68 (0.98–8.35)	0 PM	-	-	-
**1*/**2*	64.01 (56.37–71.01)	32.75 (22.62–40.43)	2.58 (0.00–7.09)	1	66.52 (59.13–75.07)	28.28 (20.13–36.11)	4.22 (1.06–7.80)
**1*/**4*	-	-	-	1.5	-	-	-
**1*/**17*	56.09 (49.23–67.34)	37.87 (23.38–43.72)	6.81 (4.27–9.11)	2	66.81 (55.54–77.35)	26.85 (17.76–38.62)	4.68 (0.98–8.35)
**2*/**2*	-	-	-	>2 UM	55.53 (49.13–67.07)	38.35 (23.88–46.00)	6.76 (2.10–8.75)
**17*/**17*		-	-	*p* ^KW^	0.041 *	0.083	0.465
**2*/**17*	-	-	-				
*p ^KW^*	0.018 *	0.057	0.243				

^KW^ Kruskal–Wallis test; ^U^ Mann–Whitney’s U test; * statistical significance (*p* < 0.05), ** statistical significance (*p* < 0.001). Showing median and interquartile ranges (p25–p75).

**Table 2 biomedicines-13-01156-t002:** Ancestry distribution percentage across different genotypes of *SLC22A1*, *SLC22A2*, *SLC22A3*, and *ABCB1* among Mexican DMT2 patients (*n* = 238).

Gen	ID	Genotype	NATAM	EUR	AFR
*SLC22A1*	rs72552763	GAT/GAT	64.04 (52.23–73.34)	30.07 (19.74–41.54)	4.68 (0.93–7.95)
		GAT/del	67.16 (55.31–77.33)	26.42 (18.83–38.77)	5.30 (1.22–8.90)
		del/del	72.10 (60.82–79.64)	24.78 (17.54–30.76)	4.19 (0.97–7.94)
		*p* ^KW^	0.089	0.090	0.597
	rs622342	A/A	64.26 (53.02–73.33)	30.07 (19.85–40.81)	4.71 (0.93–7.95)
		A/C	68.34 (55.68–77.58)	23.88 (17.63–38.71)	4.50 (1.14–8.87)
		C/C	62.94 (56.71–77.94)	29.01 (18.90–38.06)	6.04 (1.39–8.42)
		*p* ^KW^	0.430	0.225	0.474
	rs12208357	CC	65.50 (54.96–76.44)	28.40 (18.49–39.76)	4.71 (1.02–8.35)
		CT	64.40 (54.20–68.03)	27.40 (22.84–33.44)	9.38 (4.69–10.80)
		TT	-	-	-
		*p* ^U^	0.485	0.972	0.097
	rs2282143	CC	64.90 (53.94–75.92)	28.70 (19.56–40.13)	4.77 (0.99–8.66)
		CT	70.60 (62.67–78.55)	21.30 (16.33–32.62)	5.57 (1.81–7.20)
		TT	-	-	-
		*p* ^KW^	0.075	0.063	0.733
	rs594709	AA	67.20 (56.65–77.54)	26.30 (18.09–38.47)	4.68 (1.10–8.28)
		AG	63.70 (49.48–71.06)	31.30 (22.28–43.67)	4.93 (0.88–9.54)
		GG	52.60 (46.43–54.63)	41.00 (38.16–47.44)	5.62 (3.77–8.88)
		*p* ^KW^	0.008 *	0.020 *	0.854
	rs683369	CC	65.60 (55.55–76.97)	28.40 (18.32–39.08)	4.69 (1.02–8.35)
		CG	61.50 (50.35–70.72)	27.70 (22.60–42.31)	6.67 (2.90–10.32)
		GG	-	-	
		*p* ^U^	0.122	0.316	0.214
	rs628031	GG	67.20 (57.16–77.71)	26.30 (18.08–38.50)	4.59 (1.14–8.13)
		GA	63.70 (49.26–71.06)	31.30 (22.28–44.69)	5.09 (0.88–9.54)
		AA	54.30 (51.88–58.37)	35.30 (31.43–41.63)	10.40 (5.02–16.68)
		*p* ^KW^	0.022 *	0.118	0.354
*SLC22A2*	rs316019	C/C	66.52 (55.70–77.02)	27.22 (18.23–38.48)	4.80 (1.08–8.59)
		C/A	56.08 (47.69–67.88)	39.40 (25.90–48.44)	4.83 (0.97–7.57)
		A/A	-	-	-
		*p* ^U^	0.009 *	0.005 *	0.831
*SLC22A3*	*rs2076828*	C/C	68.25 (57.82–78.04)	25.83 (17.73–36.82)	4.21 (0.98–8.29)
		C/G	60.77 (50.69–70.05)	32.17 (22.17–43.88)	5.44 (2.19–9.17)
		G/G	31.09 (28.56–46.70)	62.58 (45.37–65.25)	6.38 (5.32–7.35)
		*p* ^KW^	<0.001 *	<0.001 *	0.609
*ABCB1*	rs2032582	G/G	64.08 (54.77–76.81)	30.63 (17.53–38.96)	5.00 (1.89–7.76)
		G/A	67.21 (60.81–74.26)	28.68 (21.54–35.23)	1.32 (0.13–5.74)
		A/A	-	-	-
		G/T	63.68 (51.06–72.10)	28.84 (20.91–40.59)	5.52 (2.19–9.25)
		T/T	70.66 (55.35–79.12)	23.22 (17.12–36.62)	4.62 (0.00–8.04)
		T/A	73.99 (56.67–82.36)	21.08 (14.38–38.90)	2.06 (0.75–8.16)
		*p* ^KW^	0.222	0.291	0.067
	rs1128503	C/C	63.38 (53.93–74.93)	30.15 (19.10–41.09)	3.59 (1.16–8.04)
		C/T	65.40 (53.85–77.22)	29.13 (19.70–38.87)	5.52 (1.69–8.88)
		T/T	69.40 (55.14–77.04)	24.78 (18.20–36.79)	4.00 (0.39–8.12)
		*p* ^KW^	0.291	0.380	0.297
	rs1045642	C/C	66.17 (56.33–77.83)	26.85 (17.62–38.44)	4.88 (1.24–8.41)
		C/T	63.88 (52.05–76.02)	29.10 (19.75–40.65)	4.68 (1.01–8.68)
		T/T	68.60 (54.14–74.83)	27.36 (20.15–40.85)	5.43 (1.64–8.22)
		*p* ^KW^	0.569	0.458	0.984

^KW^ Kruskal–Wallis test; ^U^ Mann–Whitney’s U test; * statistical significance (*p* < 0.05). Showing median and interquartile ranges (p25–p75).

**Table 3 biomedicines-13-01156-t003:** Correlation between ancestry proportion and allelic frequency in *CYP2C8* variants.

Allelic Variant of *CYP2C8*			
Ancestry Native-American			
	*wt*	**3*	**4*
Rho ^s^	0.250	−0.178	−0.194
*p* ^s^	<0.001 *	0.006 *	0.002 *
Ancestry EUR			
Rho ^s^	−0.207	0.162	0.149
*p* ^s^	0.001 *	0.012 *	0.023 *
Ancestry AFR			
Rho ^s^	−0.165	0.102	0.146
*p* ^s^	0.010 *	0.117	0.023 *

Rho ^s^, Spearman’s correlation coefficient; *p ^s^*, *p* value for Spearman’s correlation test; * statistical significance (*p* < 0.05).

**Table 4 biomedicines-13-01156-t004:** Correlation between ancestry proportion and allelic frequency in *CYP2C9* variants.

Allele Variant of *CYP2C9*				
Ancestry Native-American				
	*wt*	**2*	**3*	Activity score
Rho ^s^	0.135	−0.133	−0.050	0.131
*p* ^s^	0.036 *	0.039 *	0.436	0.042 *
Ancestry EUR				
Rho ^s^	−0.122	0.092	0.060	−0.121
*p* ^s^	0.059	0.154	0.352	0.062
Ancestry AFR				
Rho ^s^	−0.086	0.171	−0.037	−0.077
*p* ^s^	0.182	0.007 *	0.561	0.235

Rho ^s^, Spearman’s correlation coefficient; *p* ^s^, *p* value for Spearman’s correlation test; * statistical significance (*p* < 0.05).

**Table 5 biomedicines-13-01156-t005:** Correlation between ancestry proportion and allelic frequency in *CYP2C19* variants.

Allele Variant of *CYP2C19*					
Ancestry Native-American					
	*wt*	**2*	**4*	**17*	Activity score
Rho ^s^	0.123	0.025	-	−0.198	−0.132
*p* ^s^	0.057	0.697	-	0.002 *	0.041 *
Ancestry EUR					
Rho ^s^	−0.096	0.004	-	0.162	0.103
*p* ^s^	0.026 *	0.909	-	0.011 *	0.110
Ancestry AFR					
Rho ^s^	−0.047	−0.044	-	0.137	0.051
*p* ^s^	0.270	0.303	-	0.001 *	0.428

Rho ^s^, Spearman’s correlation coefficient; *p* ^s^, *p* value for Spearman’s correlation test; * statistical significance (*p* < 0.05).

**Table 6 biomedicines-13-01156-t006:** Correlation between ancestry proportion and allelic frequency in *SLC22A*1 and *SLC22A3* variants.

	*SLC22A1*		*SLC22A3*
	rs72552763	rs594709	rs2076828
	GAT	A	C
	Ancestry Native-American		
Rho ^s^	−0.125	0.177	0.289
*p* ^s^	0.054	0.005 *	<0.001 *
	Ancestry EUR		
Rho ^s^	0.132	−0.162	−0.258
*p* ^s^	0.041 *	0.012 *	<0.001 *
	Ancestry AFR		
Rho ^s^	−0.034	−0.035	−0.064
*p* ^s^	0.597	0.584	0.322

Rho ^s^, Spearman’s correlation coefficient; *p*
^s^, *p* value for Spearman’s correlation test; * statistical significance (*p* < 0.05).

**Table 7 biomedicines-13-01156-t007:** Global clinical and pharmacogenetic characteristics grouped by ancestry.

Variable	Native-American (*n* = 37)	Admixture (*n* = 201)	*p*
HbA1c (%) ^‖^	7.33 (6.72–9.93)	6.86 (6.10–8.58)	0.037 *
HbA1c control (<7%) ^‖^			
Yes	12 (35.29%)	86 (52.12%)	0.109
No	22 (64.70%)	79 (47.87%)	
Glucose (mg/dL)	137.93 (108.78–160.73)	129 (111–173)	0.959
Height (m)	1.46 (1.42–1.56)	1.55 (1.50–1.62)	<0.001 *
Weight (kg)	58.80 (53.40–66.30)	70.00 (62.00–80.75)	<0.001 *
BMI (kg/m^2^)	26.49 (25.26–30.72)	28.88 (25.85–33.55)	0.022 *
Diagnosis period (y)	7.00 (5.00–15.00)	6.00 (4.00–12.00)	0.214
Treatment			
Metformin	9 (28.12%)	70 (40.69%)	0.471
Metformin + glibenclamide	18 (56.25%)	84 (48.83%)	
Glibenclamide	3 (9.37%)	8 (4.65%)	
Others	2 (6.25%)	10 (5.81%)	
Metformin dose (mg/kg/day)	1.20 (0.77–1.67)	1.05 (0.64–1.50)	0.294
Triglycerides	171.00 (129.25–238.00)	178.15 (129.25–238.00)	0.779
Total cholesterol	195.00 (169.60–215.00)	199.50 (169.60–224.75)	0.558
*CYP2C8* genotype			0.275
*1/*1	36 (97.29%)	171 (85.92%)	
*1/*3	1 (2.70%)	18 (9.04%)	
*1/*4	0 (0.00%)	9 (4.52%)	
*3/*4	0 (0.00%)	1 (0.50%)	
CYP2C8 activity score			0.153
1	0 (0.00%)	1 (0.50%)	
1.5	1 (2.70%)	27 (13.56%)	
2	36 (97.29%)	171 (85.92%)	
*CYP2C9* genotype			0.710
*1/*1	35 (94.59%)	173 (86.06%)	
*1/*2	1 (2.70%)	14 (6.96%)	
*1/*3	1 (2.70%)	12 (5.97%)	
*2/*2	0 (0.00%)	1 (0.49%)	
*2/*3	0 (0.00%)	1 (0.49%)	
CYP2C9 activity score			0.552
0.5	0 (0.00%)	1 (0.49%)	
1	1 (2.70%)	13 (6.46%)	
1.5	1 (2.70%)	14 (6.96%)	
2	35 (94.59%)	173 (86.06%)	
*CYP2C19* genotype			0.252
*1/*1	31 (83.78%)	139 (69.15%)	
*1/*17	1 (2.70%)	23 (11.44%)	
*1/*2	3 (8.10%)	33 (16.41%)	
*1/*4	0 (0.00%)	1 (0.49%)	
*17/*17	0 (0.00%)	2 (0.99%)	
*2/*17	1 (2.70%)	2 (0.99%)	
*2/*2	1 (2.70%)	1 (0.49%)	
CYP2C19 activity score			0.101
PM	1 (2.70%)	1 (0.49%)	
1	3 (8.10%)	34 (16.91%)	
1.5	1 (2.70%)	2 (0.99%)	
2	31 (83.78%)	139 (69.15%)	
UM	1 (2.70%)	25 (12.43%)	
*SLC22A1* (rs72552763)			0.244
GAT/GAT	13 (35.13%)	97 (48.25%)	
GAT/del	18 (48.64%)	85 (42.28%)	
del/del	6 (16.21%)	19 (9.45)	
*SLC22A1* (rs594709)			0.059
AA	33 (89.18%)	142 (70.64%)	
AG	4 (10.81%)	55 (27.36%)	
GG	0 (0.00%)	4 (1.99%)	
*SLC22A3* (rs2076828)			0.010 *
CC	34 (91.89%)	136 (67.66%)	
CG	3 (8.10%)	59 (29.35%)	
GG	0 (0.00%)	6 (2.98%)	

* Statistical significance (*p* < 0.05). Showing median and interquartile ranges (p25–p75). ^‖^ 39 patients with no HbA1c data.

**Table 8 biomedicines-13-01156-t008:** Global genetic characteristics grouped by HbA1c control.

	HbA1c Control		
	Yes (HbA1c < 7%) *n* = 98	No (HbA1c ≥ 7%) *n* = 101	p
Native-American ancestry	64.18 (51.44–73.45)	67.16 (58.93–78.06)	0.018 *
European ancestry	29.25 (19.92–42.40)	25.16 (17.52–35.09)	0.022 *
African ancestry	5.05 (1.54–9.15)	4.19 (0.65–8.64)	0.341
Diagnosis period (y)	6 (4–10)	8 (4–15)	0.140
*CYP2C8* genotype			0.264
*1/*1	85 (87.62%)	92 (92.00%)	
*1/*3	6 (6.18%)	7 (7.00%)	
*1/*4	5 (5.15%)	1 (1.00%)	
*3/*4	1 (1.03%)	0 (0.00%)	
CYP2C8 activity score			0.426
1	1 (1.03%)	0 (0.00%)	
1.5	11 (11.34%)	8 (8.00%)	
2	85 (87.62%)	92 (92.00%)	
*CYP2C9* genotype			0.610
*1/*1	87 (88.77%)	88 (87.12%)	
*1/*2	4 (4.08%)	6 (5.94%)	
*1/*3	7 (7.14%)	5 (4.95%)	
*2/*2	0 (0.00%)	1 (0.99%)	
*2/*3	0 (0.00%)	1 (0.99%)	
CYP2C9 activity score			0.696
0.5	0 (0.00%)	1 (0.99%)	
1	7 (7.14%)	6 (5.94%)	
1.5	4 (4.08%)	6 (5.94%)	
2	87 (88.77%)	88 (87.12%)	
*CYP2C19* genotype			0.502
*1/*1	70 (71.42%)	76 (75.24%)	
*1/*17	10 (10.20%)	8 (7.92%)	
*1/*2	16 (16.32%)	12 (11.88%)	
*1/*4	0 (0.00%)	1 (0.99%)	
*17/*17	0 (0.00%)	2 (1.98%)	
*2/*17	2 (2.04%)	1 (0.99%)	
*2/*2	0 (0.00%)	1 (0.99%)	
CYP2C19 activity score			0.764
PM	0 (0.00%)	1 (0.99%)	
1	16 (16.32%)	13 (12.87%)	
1.5	2 (2.04%)	1 (0.99%)	
2	70 (71.42%)	76 (75.24%)	
UM	10 (10.20%)	10 (9.90%)	
*SLC22A1* (rs72552763)			0.730
GAT/GAT	43 (43.87%)	46 (45.54%)	
GAT/del	44 (44.89%)	47 (46.53%)	
del/del	11 (11.22%)	8 (7.92%)	
*SLC22A1* (rs594709)			0.819
AA	70 (71.42%)	74 (73.26%)	
AG	26 (26.53%)	26 (25.74%)	
GG	2 (2.04%)	1 (0.99%)	
*SLC22A3* (rs2076828)			0.347
CC	68 (69.38%)	79 (78.21%)	
CG	28 (28.57%)	21 (20.79%)	
GG	2 (2.04%)	1 (0.99%)	

* Statistical significance (*p* < 0.05). Showing median and interquartile ranges (p25–p75).

## Data Availability

The data presented in this study are available on request via the corresponding author. These data are not publicly available because the patients and researchers are bound to an agreement establishing that only the head of the study and Mexican health authorities shall have access to them, in accordance with the presidential decree of 16 April 2015, sanctioning the General Law on Transparency and Access to Public Information.

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
