# Peer review of "Molecular Ancestry Across Allelic Variants of SLC22A1, SLC22A2, SLC22A3, ABCB1, CYP2C8, CYP2C9, and CYP2C19 in Mexican-Mestizo DMT2 Patients"

_biomedicines, 2025, doi:10.3390/biomedicines13051156_

Round 1
Reviewer 1 Report
Comments and Suggestions for Authors
The manuscript entitled "Molecular ancestry across allelic variants of SLC22A1, SLC22A2, SLC22A3, ABCB1, CYP2C8, CYP2C9 and CYP2C19 in Mexican-Mestizo DMT2 patients" is carefully read and reviewed. Authors studied Native-American, European, and African ancestry distribution across 23 allelic variants of 7 genes coding pharmacokinetic proteins involved with antidiabetic drugs such as sulphonylureas (CYP2C9 and CYP2C19), thiazolidinediones (CYP2C8), and metformin (SLC22A1, SLC22A2, SLC22A3, and ABCB1). Authors provided a timely and valuable contribution to the growing field of pharmacogenomics and personalized medicine, particularly in the context of type 2 diabetes mellitus (DMT2) among the Mexican population. The study expressed a significant and underexplored area; how genetic ancestry influences pharmacogenetically relevant SNVs in DMT2. Authors emphasized personalized medicine in populations with admixed ancestry, such as Mexicans, who exhibit variable Native American, European, and African heritage.
Several issues must be considered in revision:
- The term “Mestizo” should be used with caution and clearly defined, given its socio-historical context and variability in genetic ancestry.
- Authors primarily focused on ancestry and genetics but other factors like diet, socioeconomic status, and comorbidities may influence HbA1c and should be controlled or discussed.
- Authors presented statistical associations, however, the clinical implications of specific SNVs (e.g., altered drug efficacy, adverse effects) could be more deeply discussed. Linking SNV profiles to specific antidiabetic drug responses (e.g., metformin, sulfonylureas, etc.) would enhance applicability.
- A sample of 249 patients is respectable but may limit generalizability. The results could be strengthened by replicating in larger or regionally diverse Mexican cohorts. Acknowledge please.
Author Response
Comments and Suggestions for Authors
REVIEWER 1
The manuscript entitled "Molecular ancestry across allelic variants of SLC22A1, SLC22A2, SLC22A3, ABCB1, CYP2C8, CYP2C9 and CYP2C19 in Mexican-Mestizo DMT2 patients" is carefully read and reviewed. Authors studied Native-American, European, and African ancestry distribution across 23 allelic variants of 7 genes coding pharmacokinetic proteins involved with antidiabetic drugs such as sulphonylureas (CYP2C9 and CYP2C19), thiazolidinediones (CYP2C8), and metformin (SLC22A1, SLC22A2, SLC22A3, and ABCB1). Authors provided a timely and valuable contribution to the growing field of pharmacogenomics and personalized medicine, particularly in the context of type 2 diabetes mellitus (DMT2) among the Mexican population. The study expressed a significant and underexplored area; how genetic ancestry influences pharmacogenetically relevant SNVs in DMT2. Authors emphasized personalized medicine in populations with admixed ancestry, such as Mexicans, who exhibit variable Native American, European, and African heritage.
Several issues must be considered in revision:
Q1- The term “Mestizo” should be used with caution and clearly defined, given its socio-historical context and variability in genetic ancestry.
R1- We have added the term’s conception (literally 'mixed person') to section 2.1.Study design. ‘Mestizo’ is primarily used to denote people of mixed European and non-European ancestry in the former Spanish Empire. Patient eligibility criteria were as follows: Self-proclaimed Mexican-Mestizo ancestry of at least three generations.
Rodríguez Rivera, N. S., Cuautle, P., Castillo-Nájera, F., & Molina-Guarneros, J. (2017). Identification of genetic variants in pharmacogenetic genes associated with type 2 diabetes in a Mexican-Mestizo population. Biomedical Reports, 7(1), 21-28 https://doi.org/10.3892/br.2017.921
Q2- Authors primarily focused on ancestry and genetics but other factors like diet, socioeconomic status, and comorbidities may influence HbA1c and should be controlled or discussed.
R2- We are thankful for the pertinent observation. In Mexico, medical records of public health clinics seldom register any information regarding relevant factors such as diet and socioeconomic status. This hampers analysing their impact on HbA1c control. That is also why these factors were not accounted for as inclusion criteria and also why their discussion would be impossible. A previous survey on Mexico City’s population, carried out with better infrastructure and more resources, could not account for these factors either.
Admixture in Mexico City: implications for admixture mapping of Type 2 diabetes genetic risk factors. Hum Genet (2007) 120:807–819 DOI 10.1007/s00439-006-0273-3
Q3- Authors presented statistical associations, however, the clinical implications of specific SNVs (e.g., altered drug efficacy, adverse effects) could be more deeply discussed. Linking SNV profiles to specific antidiabetic drug responses (e.g., metformin, sulfonylureas, etc.) would enhance applicability.
R3- While the study’s main objective is the assessment of ancestry distribution across different SNV genotypes in protein-coding genes involved in antidiabetic drug pharmacokinetics, we understand that a treatment-oriented stratified analysis would enhance applicability. However, while the sample size (249 individuals) might suffice to infer correlations, it is also true that more stratified analyses based on treatment would require a considerable sample size to yield an adequate statistical power and avoid spurious associations. In our sample, the metformin (n= 79) and metformin+glibenclamide (n=102) groups might be initially adequate, but the genotype sub-stratification (a minimum of 3 groups) would significantly reduce group sizes.
We have submitted a research protocol to one of the largest hospitals in Mexico, Hospital General de México Dr. Eduardo Liceaga, with the purpose of enlarging the sample’s size. This would enable adequate statistical powers to properly infer the clinical applicability of our present findings. Both the current sample size and treatment-based sub-stratification are manifest limitations of the present study. We have accepted and acknowledged them in the Discussion section.
Q4- A sample of 249 patients is respectable but may limit generalizability. The results could be strengthened by replicating in larger or regionally diverse Mexican cohorts. Acknowledge please.
R4- On the 25th of April, 2025 the Research and Research Ethics Commissions of the Hospital General de México, Dr. Eduardo Liceaga shared with us their pertaining observations upon research protocol “Polymorphic frequencies of SLC22A1/OCT1, SLC22A2/OCT2, SLC22A3/OCT3, SLC47A1/MATE1, SLC47A2/MATE2, ABCB1/MDR1, and CYP2C9 in Mexican-Mestizo DMT2 patients and their possible therapeutic associations”. These two commissions have been approved by the National Ethics Commission of Mexico (CONBIOETICA) https://www.gob.mx/salud%7Cconbioetica/articulos/comites-de-etica-en-investigacion-140023. We shall replicate the present study on a significantly broader sample of this hospital’s (the largest in Mexico) outpatient clinic once the aforementioned commissions approve its protocol.

Reviewer 2 Report
Comments and Suggestions for Authors
In this study, Ortega-Ayala et al. investigated the relationship between molecular ancestry and the frequency of pharmacogenetically relevant single nucleotide variants (SNVs) in Mexican patients with type 2 diabetes mellitus (T2DM). Their goal was to explore how genetic ancestry, Native American, European, and African, impacts antidiabetic drug response, ultimately to support the development of personalized treatment strategies.
While focusing on Mexican-Mestizo T2DM patients is contextually valuable, the study largely replicates ancestry-pharmacogenetic associations previously explored in other populations. The manuscript lacks a clear novelty claim; no new variants or mechanisms are identified, and correlations, such as between NATAM ancestry and CYP variants, have been reported before. Although the authors describe this as the "first" such study in Mexican T2DM patients, prior Mexican population studies are acknowledged, undermining this claim unless further differentiation is provided.
A clearly defined hypothesis is absent, and the study's aim appears overly broad, spanning ancestry, allele distribution, and enzyme activity, which weakens focus and clarity. Inclusion criteria are generally reasonable but ambiguously defined in places: for instance, “stable dose for at least 3 months” lacks compliance metrics, and the evaluation criteria for “sufficient medical records” are not described. Exclusion criteria are standard but do not account for potential genetic confounders, such as consanguinity or non-Mexican ancestry, which could bias ancestry-informative marker (AIM)-based analyses. Furthermore, no power calculation or justification for the sample size of 249 participants is provided.
DNA extraction and TaqMan RT-PCR genotyping are adequately described, yet crucial methodological details, such as probe specificity, genomic coverage, and validation procedures, are missing. The authors also omit discussion of genotyping error rates, replication for quality control, and handling of rare variants or failed genotype calls.
The use of 90 AIMs analyzed via ADMIXTURE is methodologically standard. However, the selection criteria for AIMs, marker density per chromosome, and validation against reference populations are not described. Additionally, the analysis includes only three ancestral populations (NATAM, EUR, AFR), overlooking the possibility of Asian or Middle Eastern admixture present in some Mexican subpopulations.
CYP2C8, CYP2C9, and CYP2C19 phenotypes were inferred from genotype data, which is an acceptable approach, but the activity score formula and specific weightings are not cited. The rationale for collapsing activity scores into categories (e.g., 1, 1.5, 2) is not provided. Some phenotype groups (gUMs or gPMs) consist of a single individual (n=1), which severely limits statistical interpretability.
The statistical analysis is weakened by the lack of adjustment for confounders and absence of multiple testing correction. Despite conducting numerous comparisons, no corrections such as Bonferroni or FDR were applied, a major methodological oversight. Additionally, no multivariate analyses were performed to adjust for variables like age, BMI, or sex, which are known to influence drug metabolism.
The manuscript’s structure, particularly in the Results section, is dense and lacks subheadings or transitions that could guide the reader through the multiple gene-specific analyses. Figures are presented in quick succession with limited commentary, reducing clarity. While the visual formats (e.g., scatterplots with regression lines) are adequate, figure legends should be more detailed to meet the expectations of a biomedical audience.
The Discussion section lacks cohesion. It begins with general population genetics, then shifts to gene-specific results without integrating the broader implications or comparing findings with previous pharmacogenetic studies in depth. Although other Latin American populations (e.g., Colombian, Brazilian, Peruvian) are referenced, there is minimal critical comparison of gene frequencies or functional consequences. Key questions remain unaddressed: What are the pharmacological implications of elevated NATAM ancestry? How do gene frequencies, not just ancestry proportions, differ from previous studies?
Finally, the clinical relevance of the ancestry-gene associations is not thoroughly discussed. The manuscript would benefit from a more explicit consideration of how these findings could influence personalized treatment decisions or therapeutic strategies in Mexican T2DM patients.
- line 409–415: numerous blank or placeholder lines – likely artifacts from conversion or formatting. Should be cleaned up.
- email addresses (“bio@gmail.com” “castillo_najera@yahoo.com.mx”), shouldn’t be in the manuscript unless required.
The references need to be updated and the journal names abbreviated according to the Index Medicus.
No graphical abstract or summary visual is provided, which could aid clarity.
Comments on the Quality of English LanguageA comprehensive review by a native English speaker or professional editor is advised. Several errors were noted in phrasing, grammar, punctuation, and awkward syntax.
- line 21: “single nucleotide allelic variants (SNVs) in protein coding genes” - should be: protein-coding genes.
- line 22: “thus fostering drug response interindividual variability” - should be: thus contributing to interindividual variability in drug response.
- line 29: “Spearman’s correlation was applied to analyse allelic and ancestry distribution.” - should be analyze (if using US English), and better phrased as to analyze the distribution of alleles and ancestry.
- line 58: “in matter of drug response” - should be: in terms of drug response.
- line 63: “glucose hepatic production” - should be: hepatic glucose production.
- line 88: “liaison between allelic variants and molecular ancestry” ; “liaison” is an odd choice. “Association” or “relationship” would be better.
- line 158: “war carried out” - should be: was carried out.
Author Response
REVIEWER 2
In this study, Ortega-Ayala et al. investigated the relationship between molecular ancestry and the frequency of pharmacogenetically relevant single nucleotide variants (SNVs) in Mexican patients with type 2 diabetes mellitus (T2DM). Their goal was to explore how genetic ancestry, Native American, European, and African, impacts antidiabetic drug response, ultimately to support the development of personalized treatment strategies.
Q1-While focusing on Mexican-Mestizo T2DM patients is contextually valuable, the study largely replicates ancestry-pharmacogenetic associations previously explored in other populations. The manuscript lacks a clear novelty claim; no new variants or mechanisms are identified, and correlations, such as between NATAM ancestry and CYP variants, have been reported before. Although the authors describe this as the "first" such study in Mexican T2DM patients, prior Mexican population studies are acknowledged, undermining this claim unless further differentiation is provided.
R1- While the correlations between ancestry and variants have indeed been studied before, the most relevant characteristic of these previous reports (1-3) is that they were performed on healthy individuals and recruitment accounts for no specific diagnosis. Our statement’s differentiation is manifest from its very heading, we have studied ancestry and pharmacogenetic biomarkers within a diagnosed DMT2 population which furthermore had been undergoing treatment for at least 3 months, the erythrocyte’s average life. This is particularly relevant for the use of HbA1c as a surrogate biomarker in the disease’s control.
- Yang HC, Chen CW, Lin YT, Chu SK. Genetic ancestry plays a central role in population pharmacogenomics. Commun Biol [Internet]. 2021 Feb 5 [cited 2025 Apr 22];4(1):171. Available from: https://www.nature.com/articles/s42003-021-01681-6
- Ramos E, Doumatey A, Elkahloun AG, Shriner D, Huang H, Chen G, et al. Pharmacogenomics, ancestry and clinical decision making for global populations. Pharmacogenomics J [Internet]. 2014 Jun [cited 2025 Apr 22];14(3):217–22. Available from: https://www.nature.com/articles/tpj201324
- Sohail M, Palma-Martínez MJ, Chong AY, Quinto-Cortés CD, Barberena-Jonas C, Medina-Muñoz SG, et al. Mexican Biobank advances population and medical genomics of diverse ancestries. Nature [Internet]. 2023 Oct 26 [cited 2025 Feb 25];622(7984):775–83. Available from: https://www.nature.com/articles/s41586-023-06560-0
Q2- A clearly defined hypothesis is absent, and the study's aim appears overly broad, spanning ancestry, allele distribution, and enzyme activity, which weakens focus and clarity. Inclusion criteria are generally reasonable but ambiguously defined in places: for instance, “stable dose for at least 3 months” lacks compliance metrics, and the evaluation criteria for “sufficient medical records” are not described. Exclusion criteria are standard but do not account for potential genetic confounders, such as consanguinity or non-Mexican ancestry, which could bias ancestry-informative marker (AIM)-based analyses. Furthermore, no power calculation or justification for the sample size of 249 participants is provided.
R2- To the best of our knowledge, exploratory studies such as this one do not require a proposed hypothesis, given that their nature is precisely raising instrumental data for any eventual hypothesis. More specifically, our study aimed to describe frequencies, ancestry proportions across different genotypes, allele correlations, and activity score. Our intent is elucidating pharmacogenetic phenomena and molecular ancestry among DMT2 Mexican patients.
Our work’s main objective is assessing relevant ancestry variables within Mexican population (NATAM, EUR, and AFR) as explained in lines 82-89.
One of the most conspicuous disadvantages of observational studies such as our own, is the absence of covariate control. This is not to diminish this design’s merit which accounts for the available evidence (4). We cannot control the medication intake of any individual within the cohort. We do have the possibility, however, of evaluating regular medical visits, periodically prescribed laboratory analyses, and (when indicated) nutrition consultations. These factors were considered compliance indicators.
We are thankful for the comments on defining a complete medical record. For the purposes of our study, we considered to be complete those records on Mexico City Healthcare Services (first-level public healthcare centres, Centro de Salud T-III Portales) rightholders containing clinical (age, gender, BMI, DMT2 diagnosis, and at least 3 months of antidiabetic drug treatment) and biochemical (including HbA1c, fasting glucose levels, triglycerides and cholesterol) data.
We are thankful for the observations on eligibility criteria. Our research protocol accounts for 3 generations of Mexican-Mestizo ancestry as an inclusion criterion [5]. Regarding consanguinity, only the first line was discarded, which means only patients’ parents, siblings, and offspring were excluded from participating.
Thank you for your observations on calculating the sample’s size. As explained in Material and methods, this study was performed in 2015 with the purpose of identifying some genetic variants such as CYP2C9. Thereby, the calculation was done as follows:
To estimate the sample’s size we used a proportion estimation formula (for the alleles in this case) over an infinite population:
|
|
|
|
|
Where
Zα2: is the value of a random variable with normal standard distribution, accumulating a probability of 1-α when alpha is 0.05 = 1.96*1.96.
q: is the estimated population parameter = 0.5.
p: value of 1-p= 0.5
d: is the confidence interval’s range = 0.06.
Since at that time (2014-2015) CYP2C9 allelic variant proportions were unknown amongst Mexican DMT2 patients, we made the following calculation where both proportions were hypothetically equal:
Sample size calculation for an unknown proportion:
In 2020 a study by Menjivar et al., reported the frequencies of some OCT1, OCT2, OCT3, MATE 1, and CYP2C9 variants in DMT2 patients undergoing metformin and glibenclamide treatment. We then performed several calculations using different allelic proportions, out of which we chose rs72552763 because it yielded the largest sample size. Thus, we based the estimation on Menjivar’s data to obtain the range of the confidence interval at 95%.
Considering p= 0.76 and q= 0.24 as previously reported by Menjivar et al., 2020, for rs72552763:
This is the sample we have calculated for allelic proportions. It corresponds to one of our DMT2 patient cohorts because we did not know the ancestry proportion across the different allelic variant genotypes and thus decided to use convenience sampling.
4.- Al Noman, A., Sarkar, O., Mita, T. M., Siddika, K., & Afrose, F. (2024). Simplifying the concept of level of evidence in lay language for all aspects of learners: In brief review. Intelligent Pharmacy, 2(2), 270–273. https://doi.org/10.1016/j.ipha.2023.11.002
[5].- Rodríguez Rivera, N. S., Cuautle, P., Castillo-Nájera, F., & Molina-Guarneros, J. (2017). Identification of genetic variants in pharmacogenetic genes associated with type 2 diabetes in a Mexican-Mestizo population. Biomedical Reports, 7(1), 21-28 https://doi.org/10.3892/br.2017.921
Q3- DNA extraction and TaqMan RT-PCR genotyping are adequately described, yet crucial methodological details, such as probe specificity, genomic coverage, and validation procedures, are missing. The authors also omit discussion of genotyping error rates, replication for quality control, and handling of rare variants or failed genotype calls.
R3- We are thankful for these detailed observations which highlight the necessity of guaranteeing the genotypic analysis’ solidity.
We used TaqMan probes, commercially developed and validated by Thermo Fisher Scientific. These are widely utilised in clinical and research contexts for the same purposes described in our manuscript. Since these assays are standardised and validated by the manufacturer, information on the probes’ specificity and their genomic coverage is not generated by our staff, but it can be consulted in the official technical documentation. To facilitate this consultation, we have included a detailed description of the assays in our supplementary material (Table S1).
For DNA extraction we used the ISO certified QIAamp DNA Blood Mini Kit (QIAGEN). Each batch of this is subjected to control facing pre-established specifications, thus guaranteeing constant and adequate quality for further applications such as PCR or sequencing, both in clinical and research contexts.
Regarding quality control:
The processing of DNA extraction was technically performed as explained in the Material and methods section. We followed the QIAGEN-indicated protocol using calibration-certified instruments, observing further quantity and quality control of the obtained DNA’s purity, integrity, and functionality, and guaranteeing their traceability at all times.
As explained in the Material and methods section, the genotyping technique entailed positive and negative controls at every assay, where heterozygous and homozygous samples of the analysed variants were involved. This strategy allowed us to guarantee result validity and reproducibility. Neither failed genotype callings, nor rare variants differing from the selected polymorphisms were detected.
It seems worth mentioning that the employed techniques and agents (including the TaqMan probes) are part of the laboratory’s habitual procedures, which have been previously performed during numerous genetic analyses, whose results have been reported in several scientific publications.
Q4-The use of 90 AIMs analyzed via ADMIXTURE is methodologically standard. However, the selection criteria for AIMs, marker density per chromosome, and validation against reference populations are not described. Additionally, the analysis includes only three ancestral populations (NATAM, EUR, AFR), overlooking the possibility of Asian or Middle Eastern admixture present in some Mexican subpopulations.
R4- The ancestry analysis was performed, as described in the Material and methods section, without supervision. We included the three parental populations to validate it. The selection of this panel was made years ago, and several published studies on Latin American populations have recurred to it (example PMIDs: 31376146, 39598523, 26391267, 35266293, 28356540). Additionally, a more detailed analysis of each SNP in this panel, including informativeness (In index) values and validation for Latin American populations, was conducted in 2012 (https://repositorio.ufmg.br/handle/1843/BUOS-95DSD7).
According to the most comprehensive study of Mexican ancestry (Sohail et al., 2023 - https://www.nature.com/articles/s41586-023-06560-0), as well as other ancestry studies, Mexican populations do not display significant percentages of ancestry components other than Native, European, and African. Therefore, selecting specific SNPs to detect other ancestries without using an array approach would be useless.
Q5- CYP2C8, CYP2C9, and CYP2C19 phenotypes were inferred from genotype data, which is an acceptable approach, but the activity score formula and specific weightings are not cited. The rationale for collapsing activity scores into categories (e.g., 1, 1.5, 2) is not provided. Some phenotype groups (gUMs or gPMs) consist of a single individual (n=1), which severely limits statistical interpretability.
R5- We have reviewed and expanded the Material and methods section (more specifically, 2.4. Genotyping procedure), to offer a more precise explanation on how the predicted phenotype was assigned based on the activity score. Moreover, the designed activity score by allele has been added to Table S1 to provide a clear and complete reference.
This clarification is relevant, given that the activity score is used numerically throughout the article. Likewise, the explanation on the phenotypic assignment of CYP2C8 (Discussion, 4.2.1. CYP2C8) has been enhanced to provide a more complete and comprehensible description.
We are thankful for the observation on the statistical limitation implied by constituting some phenotypic groups with a single individual. We are aware of this limitation and its implications for result interpretation. For this reason, in the revised manuscript we have decided to group some phenotypes with a similarly expected enzymatic activity. Concretely, individuals carrying rapid metabolising (gRMs) and ultra-rapid metabolising (gUMs) genotypes of CYP2C19 have been integrated into a single phenotypic group (UMs) to consolidate the analysis. The small number of individuals included therein is due to the low frequency of genetic variants associated with an increased enzymatic activity, such as CYP2C1917. This is noted on the manuscript, lines 579–584:
“So far, the highest CYP2C19*17 frequency has been reported among EUR (22.4%) and AFR (23.5%), whilst American individuals have reported 12.0% [40]. We have found no differences between NATAM, EUR, and AFR across the different CYP2C19 genotypes (Table 4). Moreover, the correlation analysis (Table 8) revealed a negative correlation by CYP2C19*17-NATAM, while CYP2C19*17 was positive by EUR and AFR;”
Additionally, the scarce representation of CYP2C9 within the poor metabolisers group (PMs), where only one individual was identified, reflects the low frequency of alleles CYP2C9*2 and CYP2C9*3 among the analysed population. This limitation is mentioned in lines 505–506:
“CYP2C9*2 and CYP2C9*3 frequencies are very low amongst Mexican indigenous populations [34], [35], [36], and Mexican DMT2 patients [38], [39]”.
Q6- The statistical analysis is weakened by the lack of adjustment for confounders and absence of multiple testing correction. Despite conducting numerous comparisons, no corrections such as Bonferroni or FDR were applied, a major methodological oversight. Additionally, no multivariate analyses were performed to adjust for variables like age, BMI, or sex, which are known to influence drug metabolism.
R6- We have analysed the multiple comparisons with the Bonferroni adjustment and added these results to the corresponding figures, as well as the p value of single individual groups (A).
We performed:
Multiple comparisons with an FDR and Bonferroni adjusted p value.
We added the Bonferroni adjusted p value to the corresponding figures and highlighted it in green for its prompt identification.
Based on p value adjustments, we accordingly modified the Material and methods, Results, and Discussion sections.
|
Multiple testing correction: CYP2C8 genotype and NATAM ancestry. |
||||||||||||||||||||||||||||||||||||||
|
|
|
||||||||||||||||||||||||||||||||||||
|
NATAM: Native American ancestry. FDR: False Discovery Rate (Benjamini-Hochberg). |
||||||||||||||||||||||||||||||||||||||
|
Multiple testing correction: CYP2C8 genotype and EUR ancestry. |
||||||||||||||||||||||||||||||||||||||
|
|
|
||||||||||||||||||||||||||||||||||||
|
EUR: European ancestry. FDR: False Discovery Rate (Benjamini-Hochberg). |
||||||||||||||||||||||||||||||||||||||
|
Multiple testing correction: CYP2C8 genotype and AFR ancestry. |
||||||||||||||||||||||||||||||||||||||
|
|
|
||||||||||||||||||||||||||||||||||||
|
AFR: African ancestry. FDR: False Discovery Rate (Benjamini-Hochberg). |
||||||||||||||||||||||||||||||||||||||
|
Multiple testing correction: CYP2C9 and NATAM ancestry. |
||||||||||||||||||||||||||||||||||||||||||||||||||||||||||||||||||||||||||||||||||
|
CYP2C9 genotype |
CYP2C9 activity score |
|||||||||||||||||||||||||||||||||||||||||||||||||||||||||||||||||||||||||||||||||
|
|
|
|||||||||||||||||||||||||||||||||||||||||||||||||||||||||||||||||||||||||||||||||
|
NATAM: Native American ancestry. FDR: False Discovery Rate (Benjamini-Hochberg). |
||||||||||||||||||||||||||||||||||||||||||||||||||||||||||||||||||||||||||||||||||
|
Multiple testing correction: CYP2C9 and EUR ancestry. |
||||||||||||||||||||||||||||||||||||||||||||||||||||||||||||||||||||||||||||||||||
|
CYP2C9 genotype |
CYP2C9 activity score |
|||||||||||||||||||||||||||||||||||||||||||||||||||||||||||||||||||||||||||||||||
|
|
|
|||||||||||||||||||||||||||||||||||||||||||||||||||||||||||||||||||||||||||||||||
|
EUR: European ancestry; FDR: False Discovery Rate (Benjamini-Hochberg). |
||||||||||||||||||||||||||||||||||||||||||||||||||||||||||||||||||||||||||||||||||
|
Multiple testing correction: CYP2C9 and AFR ancestry. |
||||||||||||||||||||||||||||||||||||||||||||||||||||||||||||||||||||||||||||||||||
|
CYP2C9 genotype |
CYP2C9 activity score |
|||||||||||||||||||||||||||||||||||||||||||||||||||||||||||||||||||||||||||||||||
|
|
|
|||||||||||||||||||||||||||||||||||||||||||||||||||||||||||||||||||||||||||||||||
|
AFR: African ancestry. FDR: False Discovery Rate (Benjamini-Hochberg). |
||||||||||||||||||||||||||||||||||||||||||||||||||||||||||||||||||||||||||||||||||
|
Multiple testing correction: CYP2C19 and NATAM ancestry. |
|||||||||||||||||||||||||||||||||||||||||||||||||||||||||||||||||||||||||
|
CYP2C19 genotype |
CYP2C19 activity score |
||||||||||||||||||||||||||||||||||||||||||||||||||||||||||||||||||||||||
|
|
|
||||||||||||||||||||||||||||||||||||||||||||||||||||||||||||||||||||||||
|
NATAM: Native American ancestry. FDR: False Discovery Rate (Benjamini-Hochberg). |
|||||||||||||||||||||||||||||||||||||||||||||||||||||||||||||||||||||||||
|
Multiple testing correction: CYP2C19 and EUR ancestry. |
|||||||||||||||||||||||||||||||||||||||||||||||||||||||||||||||||||||||||
|
CYP2C19 genotype |
CYP2C19 activity score |
||||||||||||||||||||||||||||||||||||||||||||||||||||||||||||||||||||||||
|
|
|
||||||||||||||||||||||||||||||||||||||||||||||||||||||||||||||||||||||||
|
EUR: European ancestry. FDR: False Discovery Rate (Benjamini-Hochberg). |
|||||||||||||||||||||||||||||||||||||||||||||||||||||||||||||||||||||||||
|
Multiple testing correction: CYP2C19 and AFR ancestry. |
|||||||||||||||||||||||||||||||||||||||||||||||||||||||||||||||||||||||||
|
CYP2C19 genotype |
CYP2C9 activity score |
||||||||||||||||||||||||||||||||||||||||||||||||||||||||||||||||||||||||
|
|
|
||||||||||||||||||||||||||||||||||||||||||||||||||||||||||||||||||||||||
|
EUR: European ancestry. FDR: False Discovery Rate (Benjamini-Hochberg). |
|||||||||||||||||||||||||||||||||||||||||||||||||||||||||||||||||||||||||
|
Multiple testing correction: rs72552762 genotype and NATAM ancestry. |
||||||||||||||||||||||||||||||||||||||
|
|
|
||||||||||||||||||||||||||||||||||||
|
NATAM: Native American ancestry. FDR: False Discovery Rate (Benjamini-Hochberg). |
||||||||||||||||||||||||||||||||||||||
|
Multiple testing correction: rs72552762 genotype and EUR ancestry. |
||||||||||||||||||||||||||||||||||||||
|
|
|
||||||||||||||||||||||||||||||||||||
|
EUR: European ancestry. FDR: False Discovery Rate (Benjamini-Hochberg). |
||||||||||||||||||||||||||||||||||||||
|
Multiple testing correction: rs72552762 genotype and AFR ancestry. |
||||||||||||||||||||||||||||||||||||||
|
|
|
||||||||||||||||||||||||||||||||||||
|
AFR : African ancestry. FDR: False Discovery Rate (Benjamini-Hochberg). |
||||||||||||||||||||||||||||||||||||||
|
Multiple testing correction: rs594709 genotype and NATAM ancestry. |
||||||||||||||||||||||||||||||||||||||
|
|
|
||||||||||||||||||||||||||||||||||||
|
NATAM: Native American ancestry. FDR: False Discovery Rate (Benjamini-Hochberg). |
||||||||||||||||||||||||||||||||||||||
|
Multiple testing correction: rs594709 genotype and EUR ancestry. |
||||||||||||||||||||||||||||||||||||||
|
|
|
||||||||||||||||||||||||||||||||||||
|
EUR: European ancestry. FDR: False Discovery Rate (Benjamini-Hochberg). |
||||||||||||||||||||||||||||||||||||||
|
Multiple testing correction: rs594709 genotype and AFR ancestry. |
||||||||||||||||||||||||||||||||||||||
|
|
|
||||||||||||||||||||||||||||||||||||
|
AFR : African ancestry. FDR: False Discovery Rate (Benjamini-Hochberg). |
||||||||||||||||||||||||||||||||||||||
|
Multiple testing correction: rs2076828 genotype and NATAM ancestry. |
||||||||||||||||||||||||||||||||||||||
|
|
|
||||||||||||||||||||||||||||||||||||
|
NATAM: Native American ancestry. FDR: False Discovery Rate (Benjamini-Hochberg). |
||||||||||||||||||||||||||||||||||||||
|
Multiple testing correction: rs2076828 genotype and EUR ancestry. |
||||||||||||||||||||||||||||||||||||||
|
|
|
||||||||||||||||||||||||||||||||||||
|
EUR: European ancestry. FDR: False Discovery Rate (Benjamini-Hochberg). |
||||||||||||||||||||||||||||||||||||||
|
Multiple testing correction: rs2076828 genotype and AFR ancestry. |
||||||||||||||||||||||||||||||||||||||
|
|
|
||||||||||||||||||||||||||||||||||||
|
AFR : African ancestry. FDR: False Discovery Rate (Benjamini-Hochberg). |
||||||||||||||||||||||||||||||||||||||
Regarding the proposal of a multivariate analysis adjusted by age, gender, and BMI, we agree with the reviewer’s observation. We have even discussed the relevance of these 3 variables for multivariate analyses in a previous publication (6). Nevertheless, in our view, the multiple adjustment according to the aforementioned variables is adequate for multiple models using clinical-response surrogate biomarkers (HbA1c, glucose, disease/no disease), whether by association or correlation; not so much for genetic conditions such as ancestry, genotypes, alleles, or activity score (B).
6.- Ortega-Ayala A, Rodríguez-Rivera NS, Andrés FD, LLerena A, Pérez-Silva E, Espinosa-Sánchez AG, et al. Pharmacogenetics of Metformin Transporters Suggests No Association with Therapeutic Inefficacy among Diabetes Type 2 Mexican Patients. Pharmaceuticals [Internet]. 2022 Jun 22 [cited 2025 Feb 24];15(7):774. Available from: https://www.mdpi.com/1424-8247/15/7/774
Q7-The manuscript’s structure, particularly in the Results section, is dense and lacks subheadings or transitions that could guide the reader through the multiple gene-specific analyses. Figures are presented in quick succession with limited commentary, reducing clarity. While the visual formats (e.g., scatterplots with regression lines) are adequate, figure legends should be more detailed to meet the expectations of a biomedical audience.
We have trimmed down the Results section. We have also added subtitles therein to offer a better reading guidance.
R7- We have added comments on the Bonferroni adjustment to the multiple comparison analyses. We have also commented on allele correlation and corresponding ancestry in each variant’s section.
We have modified figure captions. Hopefully these will live up to biomedical audiences’ expectations. We have added details such as sample, group type, group comparison variable, as well as p value adjustment, and we have now used a numerical formatting instead of asterisks.
Q8- The Discussion section lacks cohesion. It begins with general population genetics, then shifts to gene-specific results without integrating the broader implications or comparing findings with previous pharmacogenetic studies in depth. Although other Latin American populations (e.g., Colombian, Brazilian, Peruvian) are referenced, there is minimal critical comparison of gene frequencies or functional consequences. Key questions remain unaddressed: What are the pharmacological implications of elevated NATAM ancestry? How do gene frequencies, not just ancestry proportions, differ from previous studies?
R8-
- We opened the discussion with the ancestry proportion because our main objective requires its description amongst DMT2 Mexican patients. For the purpose of eventually developing plausible hypotheses NATAM, EUR, and AFR are the most relevant proportions. Therein we compared other populations from the Americas. Lines 414-418 highlight the distinct characteristics of our study’s sample. Based on this we may infer that our sample’s ancestry proportion is similar to those of healthy individuals from Ecuador, Peru, and Mexico. Consequently, we may also infer that ancestry might not be particularly relevant for diabetes risk, yet we cannot assert this because such assertion would require a different study design with the specific aim of diabetes risk identification. In our view, this section’s cohesion is based on the possibility, after describing ancestry proportions within the sample, of discussing the most relevant results of genotype-oriented ancestry distribution, beginning with the cytochromes then transitioning to transporters. Thereby we do not see the necessity for specific cytochrome/transporter order.
- We are grateful for the punctual question about pharmacological implications of NATAM’s significant preponderance. While we have modified some manuscript sections based on a positive correlation between NATAM and some genetic variants, an adequate answer to this question would require a stratified study of patients according to treatment, as well as assessing the pharmacogenetic implications of the aforementioned preponderance. We have pointed this out in lines 631-634 of the Discussion section and acknowledged it to be a manifest and important limitation of our work.
- We are grateful for the punctual question about previously reported genetic frequency differentiation beyond ancestry proportion. A truly critical comparison with other populations’ frequencies would include an analysis of those reported in the 1000 Genomes database. While these kinds of analyses have been performed and published by our research crew, we think a similar study would entail a distinct work, given that we would not have access to ancestry proportions within the 1000 Genomes sample and, even if we did, the sample would have necessitated the same methodology. The present work studied 23 SNVs, consequently a hypothetical response to this query would result in 23 tables of approximately 32 rows and 8 columns, adding up to 64 analyses for each table. That is why we consider such a critical comparison to constitute a different study, keeping in mind that our present focus is on ancestry distribution across genotypes of relevant SNVs involved in antidiabetic drug pharmacokinetics.
Q9- Finally, the clinical relevance of the ancestry-gene associations is not thoroughly discussed. The manuscript would benefit from a more explicit consideration of how these findings could influence personalized treatment decisions or therapeutic strategies in Mexican T2DM patients.
R9- Our main objective was to determine ancestry distribution across a handful of allelic variants coding proteins involved in antidiabetic drug pharmacokinetics. We realise a more explicit claim regarding these findings’ influence on personalised treatment or therapeutic strategies for Mexican DMT2 patients would benefit our work, however, assertions or conclusions seem quite premature. That is why we have noted the necessity for stratified analyses, especially considering treatment type, for example: studying patients exclusively undergoing metformin treatment and subsequently analysing only OCT variants. This kind of analysis and sub-stratification would be, in our view, derived from the present work. This seems like a promising idea, given that we have presently identified differences in HbA1c levels by categorising Mestizo and Native populations, which suggests the possibility of an ancestry-related trait leading to disease non-control (HbA1c≤7%) (Table 7). Therefore finding the correlation between a few alleles and ancestry proportions raises the possibility of analysing data with this new approach, involving multivariate analyses and even statistical modelling in the search for an eventual pattern which might explain the disease’s status over a larger sample. This sample could be stratified by treatment and variants could be analysed according to biological plausibility and in light of the present results to determine their correlation or association with HbA1c levels and/or disease status.
line 409–415: numerous blank or placeholder lines – likely artifacts from conversion or formatting. Should be cleaned up.
We have edited the manuscript accordingly.
email addresses (“bio@gmail.com” “castillo_najera@yahoo.com.mx”), shouldn’t be in the manuscript unless required.
Institutional contacts have been used instead.
The references need to be updated and the journal names abbreviated according to the Index Medicus.
We first inserted the references using Zotero, the journal’s recommended software and did not notice this. We have manually edited and updated them as well as the journals’ names according to the Index Medicus, correcting abbreviation errors.
No graphical abstract or summary visual is provided, which could aid clarity.
The graphical abstract had been submitted separately. We have now added it to the final part of the manuscript.
Comments on the Quality of English Language
A comprehensive review by a native English speaker or professional editor is advised. Several errors were noted in phrasing, grammar, punctuation, and awkward syntax.
line 21: “single nucleotide allelic variants (SNVs) in protein coding genes” - should be: protein-coding genes.
The line has been edited.
line 22: “thus fostering drug response interindividual variability” - should be: thus contributing to interindividual variability in drug response.
The line has been edited.
line 29: “Spearman’s correlation was applied to analyse allelic and ancestry distribution.” - should be analyze (if using US English), and better phrased as to analyze the distribution of alleles and ancestry.
The sentence has been rephrased. Throughout the document, British spelling has been consistently used.
line 58: “in matter of drug response” - should be: in terms of drug response.
The word has been changed.
line 63: “glucose hepatic production” - should be: hepatic glucose production.
The syntax has been edited.
line 88: “liaison between allelic variants and molecular ancestry” ; “liaison” is an odd choice. “Association” or “relationship” would be better.
The word has been changed.
line 158: “war carried out” - should be: was carried out.
The error has been corrected.

Round 2
Reviewer 2 Report
Comments and Suggestions for Authors
The authors have adequately addressed the majority of the comments and suggestions, and the manuscript has been revised accordingly. In my assessment, the manuscript is now largely suitable for publication in this journal. However, I would recommend that it undergo a final review by a native English speaker to ensure linguistic precision and clarity, thereby enhancing its overall quality.